# Gut microbe-derived trimethylamine shapes circadian rhythms through the host receptor TAAR5

Kala K Mahen[1,2,3], William J Massey[2,4], Danny Orabi[1], Amanda L Brown[1], Thomas C Jaramillo[5], Amy Burrows[1], Anthony J Horak[1,2], Sumita Dutta[1,2], Marko Mrdjen[1,2,3,4], Nour Mouannes[1,2,3], Venkateshwari Varadharajan[1,2], Lucas J Osborn[1,2,3], Xiayan Ye[1], Dante M Yarbrough[1], Treg Grubb[1], Natalie Zajczenko[1], Rachel Hohe[1,2,3], Rakhee Banerjee[1], Pranavi Linga[1], Dev Laungani[1], Adeline M Hajjar[2,6], Naseer Sangwan[2,7], Mohammed Dwidar[2,6], Jennifer A Buffa[2,6], Garth R Swanson[8], Zeneng Wang[2,6], Jonathan Mark Brown[1,2,3]*

[1]Department of Cancer Biology, Cleveland Clinic, Cleveland, United States; [2]Center for Microbiome and Human Health, Cleveland Clinic, Cleveland, United States; [3]Department of Molecular Medicine, Cleveland Clinic Lerner College of Medicine of Case Western Reserve University, Cleveland, United States; [4]Department of Inflammation and Immunity, Cleveland Clinic, Cleveland, United States; [5]Rodent Behavior Core, Cleveland Clinic, Cleveland, United States; [6]Department of Cardiovascular and Metabolic Sciences, Cleveland Clinic, Cleveland, United States; [7]Microbial Sequencing & Analytics Core Facility, Cleveland Clinic, Cleveland, United States; [8]Department of Medicine, Medical University of South Carolina, Charleston, United States

*For correspondence: brownm5@ccf.org

## eLife Assessment

This study presents an **important** finding linking the bacterial metabolite trimethylamine and its receptor to circadian rhythms and olfaction. The current evidence supporting the claims of the authors is **compelling**. This work will be of broad interest to researchers interested in nutrition, microbial metabolism, circadian rhythms, and host-microbiome interactions.

**Abstract** Elevated levels of the gut microbe-derived metabolite trimethylamine $N$-oxide (TMAO) are associated with cardiometabolic disease risk. However, the mechanism(s) linking TMAO production to human disease are incompletely understood. Initiation of the metaorganismal TMAO pathway begins when dietary choline and related metabolites are converted to trimethylamine (TMA) by gut bacteria. Gut microbe-derived TMA can then be further oxidized by host flavin-containing monooxygenases to generate TMAO. Previously, we showed that drugs lowering both TMA and TMAO protect mice against obesity via rewiring of host circadian rhythms (Schugar et al., 2022). Although most mechanistic studies in the literature have focused on the metabolic end product TMAO, here we have instead tested whether the primary metabolite TMA alters host metabolic homeostasis and circadian rhythms via trace amine-associated receptor 5 (TAAR5). Remarkably, mice lacking the host TMA receptor ($Taar5^{-/-}$) have altered circadian rhythms in gene expression, metabolic hormones, gut microbiome composition, and diverse behaviors. Also, mice genetically lacking bacterial TMA production or host TMA oxidation have altered circadian rhythms. These results provide new insights into diet–microbe–host interactions relevant to cardiometabolic disease

and implicate gut bacterial production of TMA and the host receptor that senses TMA (TAAR5) in the physiologic regulation of circadian rhythms in mice.

## Introduction

Modern investigation into human disease relies heavily on genetic and genomic approaches, and most assume that human disease is primarily driven by variation in the human genome. However, in the post-genomic era, we now understand that human genetic variation explains only a small proportion of risk for complex diseases such as obesity, diabetes, and cardiovascular disease (CVD). Instead, in many cases, environmental factors are the predominant drivers of cardiometabolic disease pathogenesis. Among all contributing environmental factors, it is clear that dietary patterns and diet-driven alterations in the gut microbiome can profoundly impact many common human diseases (*Aron-Wisnewsky and Clément, 2016*; *Nobs et al., 2020*; *Arora and Bäckhed, 2016*; *Valles-Colomer et al., 2023*). There are now many examples of diet–microbe–host interactions shaping human disease, but one of the more compelling is the reproducible link between elevated trimethylamine *N*-oxide (TMAO) levels and CVD risk (*Wang et al., 2011*; *Tang et al., 2013*; *Koeth et al., 2013*; *Zhu et al., 2016*; *Skye et al., 2018*; *Bennett et al., 2013*; *Koeth et al., 2014*; *Leng et al., 2024*). TMAO is generated by a metaorganismal (i.e. microbe and host) pathway where dietary substrates such as choline, L-carnitine, and γ-butyrobetaine are metabolized by gut microbial enzymes to generate the primary metabolite trimethylamine (TMA) (*Wang et al., 2011*; *Tang et al., 2013*; *Koeth et al., 2013*; *Zhu et al., 2016*; *Skye et al., 2018*; *Bennett et al., 2013*; *Koeth et al., 2014*; *Leng et al., 2024*). TMA is then further metabolized by the host enzyme flavin-containing monooxygenase 3 (FMO3) in the liver to produce TMAO (*Bennett et al., 2013*). Elevated TMAO levels are associated with many human diseases including diverse forms of CVD (*Wang et al., 2011*; *Tang et al., 2013*; *Koeth et al., 2013*; *Zhu et al., 2016*; *Skye et al., 2018*; *Bennett et al., 2013*; *Koeth et al., 2014*; *Leng et al., 2024*), obesity (*Schugar et al., 2017*; *Dehghan et al., 2020*), type 2 diabetes (*Miao et al., 2015*; *Steinke et al., 2020*), chronic kidney disease (CKD) (*Tang et al., 2015*; *Zixin et al., 2022*), neurodegenerative conditions including Parkinson's and Alzheimer's disease (*Kumari et al., 2020*; *Vogt et al., 2018*), and several cancers (*Xu et al., 2015*; *Banerjee et al., 2024*). Many of these disease associations have been validated in several large population meta-analyses (*Li et al., 2022*; *Heianza et al., 2017*; *Schiattarella et al., 2017*) and Mendelian randomization studies (*Jia et al., 2019*; *Zhou et al., 2023*). The emerging body of evidence supports the notion that elevated TMAO levels are causally related to cardiometabolic disease pathogenesis. In further support, therapies aimed at lowering circulating TMAO levels provide striking protection against cardiometabolic disease in animal models (*Chen et al., 2019*; *Zhu et al., 2018*; *Wang et al., 2015*; *Roberts et al., 2018*; *Organ et al., 2020*; *Schugar et al., 2022*; *Helsley et al., 2022*; *Zhang et al., 2021*; *Gupta et al., 2020*). Several independent studies have now shown that inhibition of either the host TMAO-producing enzyme FMO3 or the microbial TMA-producing enzyme CutC protects against diet-induced atherosclerosis (*Miao et al., 2015*; *Wang et al., 2015*), heart failure (*Organ et al., 2020*), thrombosis (*Zhu et al., 2018*; *Roberts et al., 2018*), obesity (*Schugar et al., 2017*; *Schugar et al., 2022*), liver disease (*Helsley et al., 2022*), insulin resistance (*Schugar et al., 2017*; *Schugar et al., 2022*), CKD (*Zhang et al., 2021*; *Gupta et al., 2020*), and abdominal aortic aneurysm (*Benson et al., 2023*).

Even though TMAO-lowering therapies are very effective in preclinical animal models, the underlying mechanism(s) linking the metaorganismal production of TMAO to cardiometabolic disease pathogenesis are still incompletely understood. Mechanistic understanding has been hampered by the fact that it is hard to disentangle the pleiotropic effects of dietary substrates (choline, carnitine, γ-butyrobetaine, trimethyllysine, etc.), bacterial production of TMA, and host-driven conversion of TMA to TMAO. At this point, the vast majority of studies have focused on the end product of this pathway, TMAO, but it is equally plausible that the primary metabolite, TMA, may play some role in microbe–host crosstalk related to human disease. There is some evidence that TMAO can promote inflammatory processes via activation of the nucleotide-binding domain, leucine-rich-containing family, pyrin domain-containing-3 (NLRP3) inflammasome, and nuclear factor κB (NFκB) (*Seldin et al., 2016*; *Sun et al., 2016*; *Chen et al., 2017*; *Zhang et al., 2020*). TMAO can also activate the endoplasmic reticulum (ER) stress kinase PERK (EIF2AK3) in hepatocytes to promote metabolic disturbance (*Chen et al., 2019*). In parallel, TMAO promotes stimulus-dependent calcium release in platelets

to promote thrombosis (*Zhu et al., 2016*). Although the end product of the pathway TMAO clearly impacts cell signaling in the host to impact cardiometabolic disease, these TMAO-driven mechanisms do not fully explain how elevated TMAO levels contribute to so many diverse diseases in humans. In addition to TMAO-driven signaling mechanisms, we recently reported that major components of the TMAO pathway (choline, TMA, FMO3, and TAAR5) oscillate in a highly circadian fashion (*Schugar et al., 2022*). Furthermore, gut microbe-targeted drugs that selectively block TMA production alter host circadian rhythms in the gut microbiome and host phospholipid metabolism (*Schugar et al., 2022*). It is important to note that disruption of the circadian clock is a common hallmark of almost all diseases where TMAO levels are elevated (*Sulli et al., 2018*; *Bolshette et al., 2023*; *Zheng et al., 2020*; *Bishehsari et al., 2020*). To follow up on the potential links between the TMAO pathway and host circadian disruption, here we have used genetic knockout approaches at the level of host sensing of TMA (i.e. *Taar5⁻/⁻*), gut microbial TMA production (i.e. *cutC*-null microbial communities), and host TMA oxidation (i.e. *Fmo3⁻/⁻*). Results here further bolster the concept that TMA production and associated TAAR5 activation shapes host circadian rhythms.

## Results

### Mice lacking the TMA receptor TAAR5 have altered circadian rhythms in metabolic homeostasis and innate olfactory-related behaviors

We recently demonstrated that drugs blocking gut microbial TMA production protect against obesity via rewiring circadian rhythms in the gut microbiome, liver, white adipose tissue (WAT), and skeletal muscle (*Schugar et al., 2022*). Furthermore, we also showed that blocking bacterial TMA production elicited unexpected alterations in olfactory perception of diverse odorant stimuli (*Massey et al., 2023*). Therefore, we have followed up here to further interrogate circadian rhythms in the gut microbiome, liver, WAT, skeletal muscle, and olfactory bulb in mice lacking the only known host G-protein-coupled receptor (GPCR) that senses TMA known as TAAR5 (*Wallrabenstein et al., 2013*; *Li et al., 2013*). In agreement with our previous report showing that *Taar5* mRNA is expressed in a circadian manner in skeletal muscle (*Schugar et al., 2022*), *LacZ* reporter expression oscillates with peak expression in the dark cycle in skeletal muscle (*Figure 1A*). However, unlike the striking impact that choline TMA lyase inhibitors have on the core circadian clock machinery in skeletal muscle (*Schugar et al., 2022*), mice lacking the TMA receptor (*Taar5⁻/⁻*) also have largely unaltered circadian gene expression in skeletal muscle, with only nuclear receptor subfamily 1 group D member 1 (*Nr1d1*, *Rev-Erbα*) showing a modest yet significant delay in acrophase (*Figure 1A*; *Supplementary file 1*). Instead, *Taar5⁻/⁻* mice have alterations in the expression of key circadian genes including basic helix-loop-helix ARNT like 1 (*Bmal1*), clock circadian regulator (*Clock*), *Nr1d1*, cryptochrome 1 (*Cry1*), and period 2 (*Per2*) in the olfactory bulb (*Figure 1B*; *Supplementary file 1*). *Taar5⁻/⁻* mice exhibited increased mesor for *Bmal1*, *Clock*, and *Nr1d1* compared to *Taar5⁺/⁺* controls in the olfactory bulb (*Figure 1B*; *Supplementary file 1*). Whereas, *Taar5⁻/⁻* mice exhibited advanced acrophase in *Cry1* and *Per2* compared to *Taar5⁺/⁺* controls in the olfactory bulb (*Figure 1B*; *Supplementary file 1*). There is also some reorganization of circadian gene expression in the liver (*Figure 1C*) and gonadal WAT (*Figure 1D*; *Supplementary file 1*), albeit modest. In the liver, *Taar5⁻/⁻* mice had normal circadian gene expression when compared to *Taar5⁺/⁺* controls (*Figure 1C*; *Supplementary file 1*). In gonadal WAT, the amplitude of *Bmal1* was increased and acrophase of *Bmal1* was delayed in *Taar5⁻/⁻* mice, whereas only the acrophase of Per2 was advanced in *Taar5⁻/⁻* mice compared to *Taar5⁺/⁺* controls (*Figure 1D*; *Supplementary file 1*). Given the key role that the TMAO pathway plays in suppressing the beiging of WAT (*Schugar et al., 2017*), we also examined the expression of PR/SET domain 16 (Prdm16) and uncoupling protein 1 (Ucp1). *Taar5⁻/⁻* mice had increased mesor and advanced acrophase of Prdm16, yet no difference was detected in Ucp1, compared to *Taar5⁺/⁺* controls (*Figure 1D*; *Supplementary file 1*). *Taar5⁻/⁻* mice have unaltered oscillations in body weight (*Figure 1—figure supplement 1*; *Supplementary file 1*).

We next examined circadian oscillations in circulating metabolite, hormone, and cytokine levels in *Taar5⁺/⁺* and *Taar5⁻/⁻* mice (*Figure 1—figure supplement 2*; *Supplementary file 1*). When we measured substrates for gut microbial TMA production, we found that *Taar5⁻/⁻* mice had normal oscillations in choline and ʟ-carnitine (*Figure 1—figure supplement 2A*; *Supplementary file 1*). However, there was a modest yet significant advance in the acrophase of plasma γ-butyrobetaine in *Taar5⁻/⁻* mice (*Figure 1—figure supplement 2A*; *Supplementary file 1*). *Taar5⁻/⁻* mice also had slightly increased

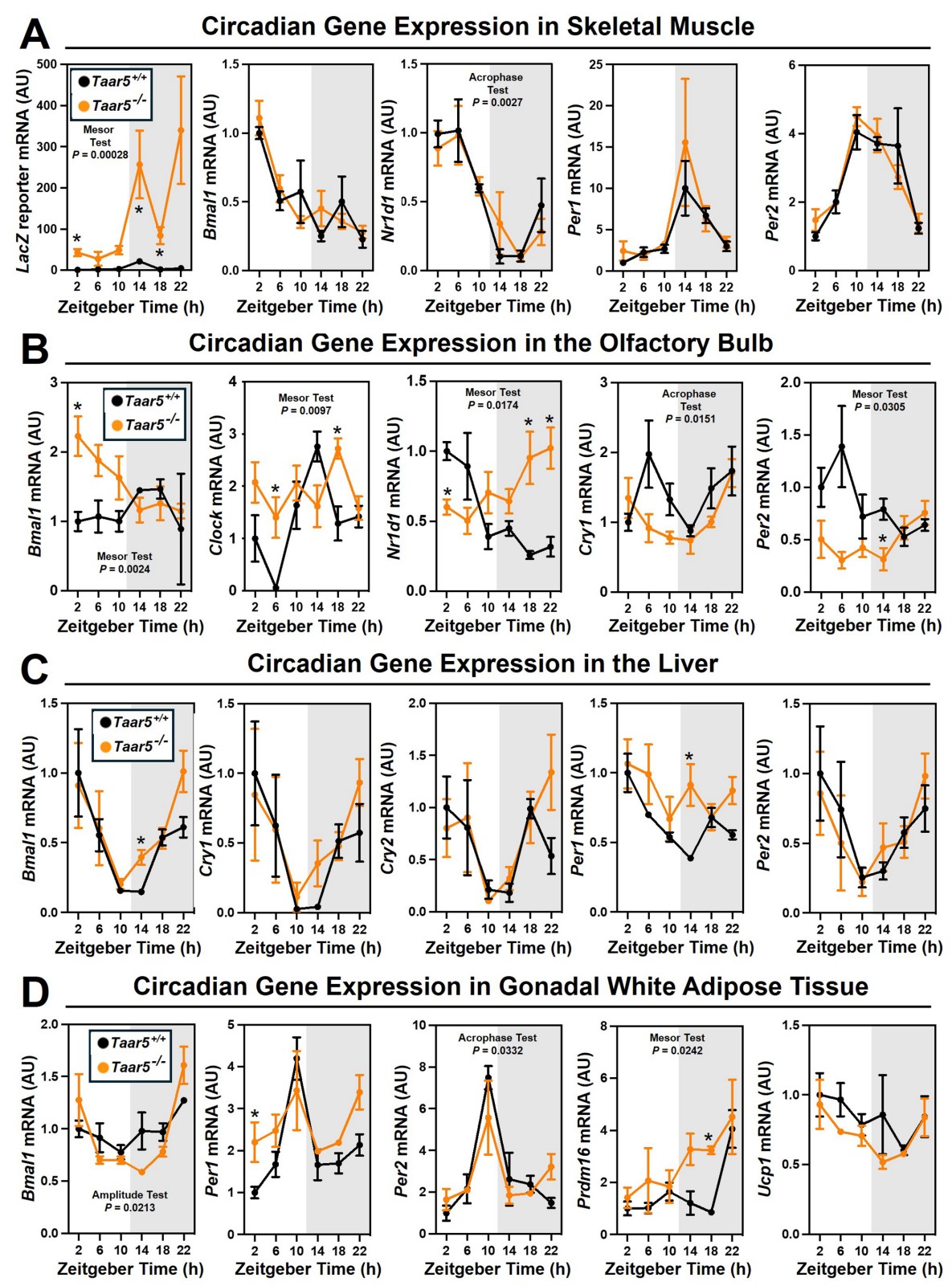

**Figure 1.** The host trimethylamine receptor TAAR5 shapes tissue-specific circadian oscillations. Male chow-fed wild-type (*Taar5*^+/+^) or mice lacking the TMA receptor (*Taar5*^-/-^) were necropsied at 4-hr intervals to collect tissues including skeletal muscle (**A**), olfactory bulb (**B**), liver (**C**), or gonadal white adipose tissue (**D**). The relative gene expression for circadian (*Bmal1*, *Clock*, *Nr1d1*, *Cry1*, and *Per2*) and metabolism (*Prdm16* and *Ucp1*) related genes was quantified by qPCR using the ΔΔ-CT method. Data shown represent the means ± SEM for *n* = 3–6 individual mice per group. Group differences

*Figure 1 continued on next page*

*Figure 1 continued*

were determined using cosinor analyses, and p-values are provided where there were statistically significant differences between *Taar5*[+/+] and *Taar5*[-/-] mice. The complete cosinor statistical analysis for circadian data can be found in **Supplementary file 1**. *Significant differences between *Taar5*[+/+] and *Taar5*[-/-] mice by Student's *t*-tests within each ZT time point (p < 0.05).

The online version of this article includes the following figure supplement(s) for figure 1:

**Figure supplement 1.** Body weights across a 24-hr period in chow-fed male *Taar5*[+/+] and *Taar5*[-/-] mice.

**Figure supplement 2.** The host trimethylamine receptor TAAR5 shapes circadian oscillations in circulating hormones and cytokines.

levels of TMA at ZT22, when compared to *Taar5*[+/+] controls (**Figure 1—figure supplement 2A**; **Supplementary file 1**). However, TMA and TMAO levels were not significantly altered in *Taar5*[-/-] mice (**Figure 1—figure supplement 2A**; **Supplementary file 1**). Interestingly, *Taar5*[-/-] mice had altered rhythmic levels in some but not all host metabolic hormones. Although the circadian oscillations of plasma insulin and C-peptide were not significantly altered, *Taar5*[-/-] mice had a delay in acrophase of plasma glucagon, yet advanced acrophase of glucagon-like peptide 1 (GLP-1), compared to *Taar5*[+/+] controls (**Figure 1—figure supplement 2B**; **Supplementary file 1**). *Taar5*[-/-] mice also exhibited increased mesor for plasma leptin levels compared to wild-type controls (**Figure 1—figure supplement 2B**; **Supplementary file 1**). *Taar5*[-/-] mice also had modest differences in circulating cytokines, including a decrease in the mesor of monocyte chemoattractant 1 (MCP-1) and tumor necrosis factor α (TNFα) (**Figure 1—figure supplement 2B**; **Supplementary file 1**). Collectively, these data demonstrate that *Taar5*[-/-] mice have altered circadian-related gene expression that is most apparent in the olfactory bulb (**Figure 1**; **Supplementary file 1**), and abnormal circadian oscillations in some but not all circulating hormones and cytokines (**Figure 1—figure supplement 2**; **Supplementary file 1**).

We next set out to comprehensively analyze the circadian rhythms in behavioral phenotypes in mice lacking the TMA receptor TAAR5. In this line of investigation, we took a very broad approach to examine impacts of *Taar5* deficiency on metabolic, cognitive, motor, anxiolytic, social, olfactory, and innate behaviors. The main rationale behind this in-depth investigation was to allow for comparison to our recent work showing that pharmacologic blockade of the production of the TAAR5 ligand TMA (using choline TMA lyase inhibitors) produced clear metabolic, innate, and olfactory-related social behavioral phenotypes, but did not dramatically impact aspects of cognition, motor, or anxiolytic behaviors (**Schugar et al., 2022**; **Massey et al., 2023**). Also, it is important to note other groups have recently shown that TAAR5 activation with non-TMA ligands or genetic deletion of *Taar5* results in clear olfactory (**Li et al., 2013**; **Liberles, 2015**; **Freyberg and Saavedra, 2020**; **Espinoza et al., 2020**), anxiolytic (**Espinoza et al., 2020**), cognitive (**Maggi et al., 2022**), and sensorimotor (**Aleksandrov et al., 2019**; **Kalinina et al., 2021**) behavioral abnormalities. Here, it was our main goal to identify whether any behavioral phenotypes that were consistently seen in mice lacking bacterial TMA production (**Schugar et al., 2022**; **Massey et al., 2023**) or host TMA sensing by TAAR5 (studied here) are time-of-day dependent indicating circadian inputs. Given the consistent alterations in innate and olfactory-related phenotypes seen in both mice lacking bacterial TMA production (**Schugar et al., 2022**; **Massey et al., 2023**) and mice lacking host TMA sensing by TAAR5 (**Wallrabenstein et al., 2013**; **Li et al., 2013**; **Liberles, 2015**; **Freyberg and Saavedra, 2020**; **Espinoza et al., 2020**; **Maggi et al., 2022**; **Aleksandrov et al., 2019**; **Kalinina et al., 2021**), we next followed up to perform select innate and olfactory-related behavioral tests in mice at defined circadian time points (**Figure 2**).

To study the circadian presentation of phenotype, we carefully controlled the time window of testing for either the olfactory cookie test or marble burying test, both of which have been shown to be altered in mice lacking bacterial TMA synthesis (**Massey et al., 2023**; **Romano et al., 2017**). When the olfactory cookie test was performed during the mid-light cycle (ZT5–ZT7), the latency to find the buried cookie was significantly increased in male *Taar5*[-/-] mice compared to *Taar5*[+/+] controls (**Figure 2A**). However, when the same mice performed the olfactory cookie test at the dark-to-light phase transition (ZT23–ZT1), or at the light-to-dark phase transition (ZT13–ZT15), there were no significant differences between *Taar5*[+/+] and *Taar5*[-/-] mice (**Figure 2A**). When subjected to the marble burying test, only female *Taar5*[-/-] mice buried significantly more marbles than female wild-type controls only at ZT5–ZT7, but this was not apparent at other ZT time points (**Figure 2B**). Collectively, these data demonstrate that *Taar5*[-/-] mice exhibit highly gender-specific alterations in innate and olfactory

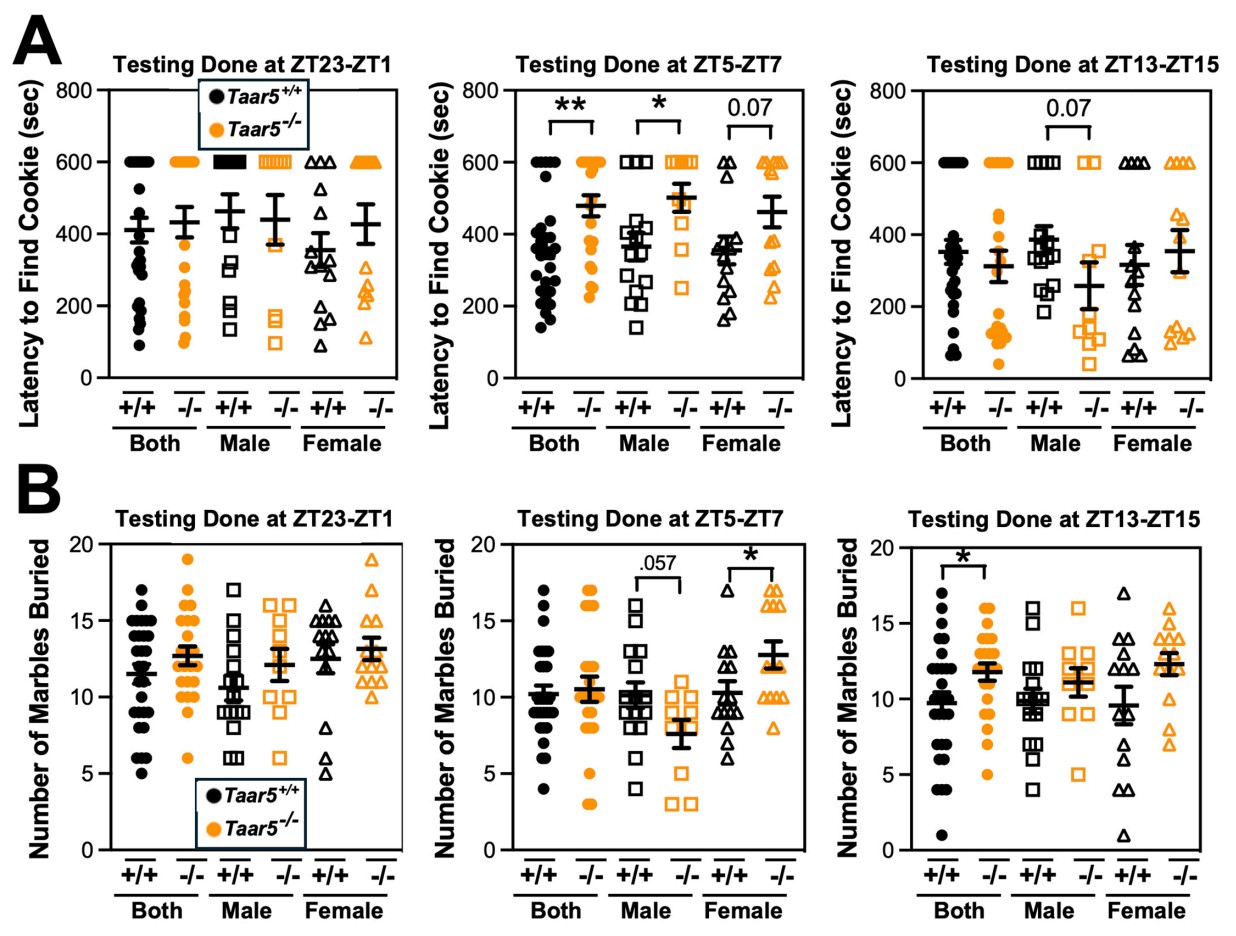

**Figure 2.** Mice lacking the host TMA receptor TAAR5 have altered olfactory and repetitive behaviors only at specific circadian time points. Male or female wild-type mice (*Taar5*⁺/⁺) or mice lacking the TMA receptor (*Taar5*⁻/⁻) were subjected to the olfactory cookie test (**A**) or the marble burying test (**B**). To examine circadian alterations in behavior, these tests were done in either the dark-light phase transition (ZT23–ZT1), mid light cycle (ZT5–ZT7), or early dark cycle (ZT13–ZT15). Data represent the mean ± SEM from $n$ = 10–15 per group when male and female are separated ($n$ = 25–27 when both sexes are combined), and statistically significant difference between *Taar5*⁺/⁺ and *Taar5*⁻/⁻ mice are denoted by *p < 0.05 and **p < 0.01.

The online version of this article includes the following figure supplement(s) for figure 2:

**Figure supplement 1.** Olfactory discrimination of several odor stimuli is unaltered in *Taar5*-deficient mice.

**Figure supplement 2.** *Taar5*-deficient mice exhibit specific alterations in social behaviors.

**Figure supplement 3.** *Taar5*-deficient mice exhibit specific alterations in innate behavioral responses.

**Figure supplement 4.** Impact of *Taar5* deficiency on cognitive, depression, and anxiety-like behaviors.

**Figure supplement 5.** Impact of *Taar5* deficiency on Morris water maze performance.

**Figure supplement 6.** Impact of *Taar5* deficiency on systemic energy metabolism and gene expression in brown adipose tissue (BAT).

behaviors, and these behavior phenotypes are only apparent at certain periods within the light cycle (*Figure 2* and S2).

Although *Taar5*⁻/⁻ mice showed time-dependent alterations in the olfactory cookie test, olfactory discrimination toward other diverse single stimuli such as banana, corn oil, almond, water, or social cues was not significantly altered (*Figure 2—figure supplement 1*). When we subjected *Taar5*⁻/⁻ mice to a battery of social behavioral tests, there were test-specific alterations that occurred in a sexually dimorphic manner. All mouse groups (*Taar5*⁺/⁺, *Taar5*⁻/⁻, male and female) displayed no initial chamber bias in the initial trial of the three-chamber test (*Figure 2—figure supplement 2A*). In the three-chamber preference test, only male *Taar5*⁻/⁻ mice showed no preference between an inanimate object and a social stimuli interaction (*Figure 2—figure supplement 2B*). When subjected to the three-chamber social novelty test box, both male and female *Taar5*⁻/⁻ mice showed no preference between

the novel and familiar stimuli (*Figure 2—figure supplement 2C*). In the social interaction with a juvenile mouse test, *Taar5*[-/-] females showed no significant difference in interaction time between the initial interaction trial and the recognition trial 4 days later (*Figure 2—figure supplement 2D*). In addition to alterations in social interactions, *Taar5*[-/-] mice also showed sexually dimorphic alterations in several other innate behavioral tests (*Figure 2—figure supplement 3*). Female *Taar5*[-/-] mice exhibited a significantly higher startle response at 90, 100, 110, and 120 decibels (*Figure 2—figure supplement 3A*), and significantly weaker forelimb grip strength (*Figure 2—figure supplement 3B*) compared to *Taar5*[+/+] controls. When both sexes are combined, there is a significant increase in the latency to withdraw during the hotplate sensitivity test in *Taar5*[-/-] mice compared to controls (*Figure 2—figure supplement 3C*). Furthermore, both male and female *Taar5*[-/-] mice have slightly increased latency to fall during the rotarod test (*Figure 2—figure supplement 3D*). Also, female but not male *Taar5*[-/-] mice exhibit reduced nest building compared to *Taar5*[+/+] mice (*Figure 2—figure supplement 3E*).

We next comprehensively examined cognitive, depression, and anxiety-like behaviors in *Taar5*[+/+] and *Taar5*[-/-] mice (*Figure 2—figure supplements 4 and 5*). Both male and female *Taar5*[-/-] mice performed similarly to *Taar5*[+/+] controls in the open field, elevated plus maze, and Y-maze tests (*Figure 2—figure supplement 4B–D*). However, female, but not male, *Taar5*[-/-] mice showed significantly reduced freezing compared to *Taar5*[+/+] controls in the cued fear conditioning test (*Figure 2—figure supplement 4A*). When subjected to the Morris water maze, there were only minor alterations found in *Taar5*[-/-] mice. All mice showed similar latency to the platform. Male *Taar5*[-/-] mice showed increased distance traveled compared to wild-type controls, yet females were more similar to *Taar5*[+/+] controls (*Figure 2—figure supplement 5*). However, female *Taar5*[-/-] mice showed increased velocity compared to *Taar5*[+/+] mice during the last 3 days of testing (*Figure 2—figure supplement 5*). Collectively, the impact of *Taar5* deficiency on cognitive, depression, and anxiety-like behaviors was very modest.

Given the TMAO pathway has been linked to the beiging of WAT and energy expenditure (*Schugar et al., 2017*), we next examined circadian rhythms in energy metabolism during a cold challenge and gene expression in thermogenic brown adipose tissue (BAT) in *Taar5*[-/-] mice (*Figure 2—figure supplement 6*). Although male *Taar5*[-/-] mice showed unaltered oxygen consumption at thermoneutrality (30°C), room temperature (22°C), and during cold (4°C) exposure, female *Taar5*[-/-] mice had significantly elevated oxygen consumption that appeared most significant during the light cycle periods (*Figure 2—figure supplement 6A*). To follow up, we collected BAT from *Taar5*[+/+] and *Taar5*[-/-] mice at ZT2 (early light cycle) and ZT14 (early dark cycle) to examine potential alterations in circadian gene expression. Both male and female *Taar5*[-/-] mice showed marked upregulation of *Bmal1*, yet other than increased *Per1* expression at ZT14, all other circadian genes were largely unaltered in *Taar5*[-/-] mice (*Figure 2—figure supplement 6B*). Taken together, all behavioral data presented here show that mice lacking *Taar5* have select sexually dimorphic alterations in olfactory, innate, social, and metabolic phenotypes.

## Host TAAR5 regulates the circadian rhythmicity of the gut microbiome

Bi-directional microbe–host communication is required for homeostatic control of chronobiology in the metaorganism (*Mukherji et al., 2013*; *Asher and Sassone-Corsi, 2015*; *Thaiss et al., 2014*; *Liang et al., 2015*; *Choi et al., 2021*). Our data strongly suggest that gut microbe-derived TMA can shape host circadian rhythms in metabolic homeostasis and behavior (*Schugar et al., 2022*; *Massey et al., 2023*), but we also wanted to test whether host TAAR5 may reciprocally regulate circadian rhythms of the gut microbiome (*Thaiss et al., 2014*; *Liang et al., 2015*). The rationale for studying alterations in the gut microbiome stems from the fact that several previous studies show that manipulating the TMAO pathway at different levels (diet, microbe, and host) strongly alters gut microbiota in unexpected ways (*Koeth et al., 2013*; *Koeth et al., 2014*; *Roberts et al., 2018*; *Schugar et al., 2022*). For example, provision of dietary substrates (i.e. choline or L-carnitine) or exogenous TMAO itself can reorganize gut microbial communities in mice (*Koeth et al., 2013*; *Koeth et al., 2014*). Also, small molecule enzyme inhibitors blocking the bacterial production of TMA profoundly alter the cecal microbiome (*Roberts et al., 2018*; *Schugar et al., 2022*), and some of the anti-obesity and circadian rhythm-altering effects can be transmitted by cecal microbial transplantation (*Schugar et al., 2022*). Therefore, we examined the circadian oscillation in the cecal microbiome in *Taar5*[+/+] and *Taar5*[-/-] mice over a 24-hr period and found there were clear alterations that were time of day dependent (*Figure 3*, *Figure 3—figure supplements 1 and 2*; *Supplementary file 1*). At the phylum level, wild-type mice

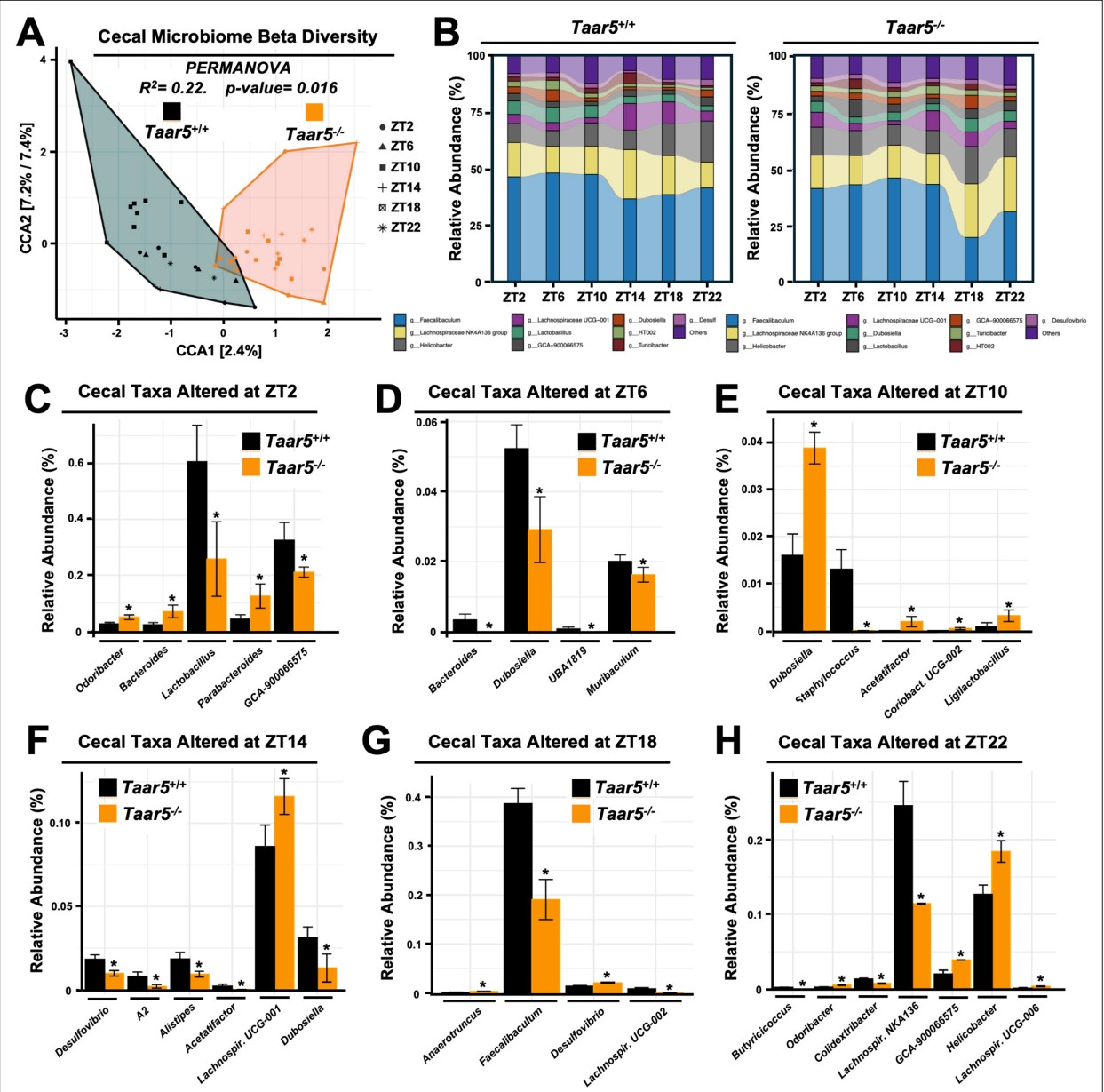

**Figure 3.** The trimethylamine receptor TAAR5 shapes circadian oscillations in the gut microbiome. Male chow-fed wild-type (*Taar5⁺/⁺*) or mice lacking the TMA receptor (*Taar5⁻/⁻*) were necropsied at 4-hr intervals to collect cecum for microbiome composition analyses via sequencing the V4 region of the 16S rRNA (genus level changes are shown). (**A**) Canonical correspondence analysis (CCA) based beta diversity analyses show distinct microbiome compositions in *Taar5⁺/⁺* and *Taar5⁻/⁻* mice. Statistical significance and beta dispersion were estimated using PERMANOVA. (**B**) The relative abundance of cecal microbiota in *Taar5⁺/⁺* and *Taar5⁻/⁻* mice. Significantly altered cecal microbial genera in *Taar5⁺/⁺* and *Taar5⁻/⁻* mice are shown at ZT2 (**C**), ZT6 (**D**), ZT10 (**E**), ZT14 (**F**), ZT18 (**G**), and ZT22 (**H**). ASVs that were significantly different in abundance (MetagenomeSeq with Benjamini–Hochberg false discovery rate (FDR) multiple test correction, adjusted p < 0.01). Data shown represent the means ± SD for n = 3–6 individual mice per group. Group differences were determined using ANOVA with Benjamini–Hochberg FDR multiple test correction, *adjusted p < 0.01.

The online version of this article includes the following figure supplement(s) for figure 3:

**Figure supplement 1.** TAAR5-deficient mice have altered circadian oscillations in the gut microbiome at the phylum level.

**Figure supplement 2.** TAAR5-deficient mice have altered circadian oscillations in the gut microbiome.

showed a modest decrease in the light cycle (i.e. ZT2–ZT10), and progressive loss of *Firmicutes* over the dark cycle (i.e. from ZT14–ZT22). In contrast, *Taar5⁻/⁻* mice showed a reciprocal increase during the light cycle and more modest dip during the light cycle in Firmicutes (***Figure 3—figure supplement 1***; ***Supplementary file 1***). When we examined microbiome alterations at the genus level, there were

much clearer alterations in specific bacteria (*Figure 3*; *Figure 3—figure supplement 2*; *Supplementary file 1*). In particular, the acrophase of several *Lachnospiraceae*, *Odoribacter*, and *Dubosiella* genera is altered in *Taar5*[-/-] mice when compared to wild-type controls (*Figure 3*; *Figure 3—figure supplement 2*; *Supplementary file 1*). Although there were many microbiome alterations, *Taar5*[-/-] mice showed delayed acrophase for *Lachnospiraceae UCG-002*, *Lachnospiraceae NK4A136*, *Desulfovibrio*, *Bacteroides*, *Dubosiella*, *Colidextribacter*, *Alistipes*, *Turicibacter*, *Muribaculum*, *Helicobacter*, and *Parabacteroides*, when compared to *Taar5*[+/+] mice (*Figure 3—figure supplement 2*; *Supplementary file 1*). Other genera such as *Lachnospiraceae UCG-006*, *Odoribacter*, *Butyricicoccus*, *Coriobacteriaceae UCG-002*, *Acetatifactor*, and *A2* showed advanced acrophase in *Taar5*[-/-] mice (*Figure 3—figure supplement 2*; *Supplementary file 1*). It is important to note that previous independent studies have also shown that either blocking bacterial TMA synthesis (*Schugar et al., 2022*) or FMO3-driven TMA oxidation (*Zhu et al., 2018*) can also strongly reorganize the gut microbiome in mice. Although more work is needed to fully understand the underlying mechanisms, it is clear that both gut microbe-driven TMA production and host sensing of TMA by TAAR5 (*Figure 3*; *Figure 3—figure supplement 2*; *Supplementary file 1*) can strongly impact circadian oscillations of the gut microbiome.

## Mice genetically lacking either gut microbial TMA production or host-driven TMA oxidation have altered circadian rhythms

To confirm and extend the idea that gut microbe-derived TMA can shape host circadian rhythms, we next performed experiments where we genetically deleted either gut microbial TMA synthesis or host-driven TMA oxidation. First, we used gnotobiotic mice engrafted with a defined microbial community with or without genetic deletion of the choline TMA lyase CutC (i.e. *Clostridium sporogenes* wild-type versus Δ*cutC*) to understand the ability of gut microbe-derived TMA to alter host circadian rhythms (*Figure 4*; *Figure 4—figure supplement 1*; *Figure 4—figure supplement 2*). Circadian oscillations in plasma TMA and TMAO that peak in the early dark cycle are only detectable in the group colonized with *C. sporogenes* (WT), but not Δ*cutC C. sporogenes* (*Figure 4A*). Other TMAO pathway-related metabolites, including choline, *L*-carnitine, γ-butyrobetaine, and betaine, also exhibited modest alterations in circadian rhythms (*Figure 4A*). Mice lacking choline TMA lyase activity showed modestly reduced mesor in plasma choline and advanced acrophase in both plasma betaine and γ-butyrobetaine (*Figure 4A*; *Supplementary file 1*). In a similar manner to *Taar5*[-/-] mice (*Figure 1*), mice colonized with Δ*cutC C. sporogenes* show alterations in the expression of some but not all circadian genes in the olfactory bulb (*Figure 4B*; *Supplementary file 1*). Mice colonized with Δ*cutC C. sporogenes* had significantly advanced acrophase for *Bmal1*, yet delayed acrophase for Clock, in the olfactory bulb when compared to mice colonized with the control community (*Figure 4B*; *Supplementary file 1*). However, the oscillatory pattern of other circadian genes was not significantly altered. When we examined circulating hormones and cytokines, there were clear alterations in the circadian oscillation in mice genetically lacking choline TMA lyase activity (*Figure 4C*; *Supplementary file 1*). Mice colonized with the Δ*cutC C. sporogenes* community exhibited delayed acrophase for plasma insulin and interleukins 2 (IL-2) and 33 (IL-33) when compared to mice harboring the wild-type community that could produce TMA (*Figure 4C*; *Supplementary file 1*). Mice lacking choline TMA lyase activity also had an advanced acrophase for GLP-1, leptin, and IL-1β (*Figure 2C*; *Supplementary file 1*). It is interesting to note that we examined the cecal abundance of the five bacterial strains in our defined community. There were clear alterations that were time of day dependent (*Figure 4—figure supplement 1*). Four of the bacterial strains (*C. sporogenes*, *B. theta*, *B. caccae*, and *B. ovatus*) showed clear circadian oscillations when a zero amplitude test was performed in mice colonized with wild-type community. However, the community lacking *cutC* lost all significant circadian oscillation of these same four bacterial strains (*Figure 4—figure supplement 1*; *Supplementary file 1*), showing that bacterial production of TMA can broadly impact both microbe–microbe and microbe–host interactions in circadian rhythms.

To follow up on our previous findings that FMO3 suppresses both beige and brown fat-induced cold-induced thermogenesis (*Schugar et al., 2017*), we also wanted to examine circadian patterns in subscapular BAT of Δ*cutC C. sporogenes*-colonized mice (*Figure 4—figure supplement 2*; *Supplementary file 1*). Although there were no statistically significant alterations in circadian gene expression in the BAT isolated from Δ*cutC C. sporogenes*-colonized mice, the acrophase of phosphatidylethanolamine methyltransferase (*Pemt*) was delayed in mice colonized with the Δ*cutC* community compared

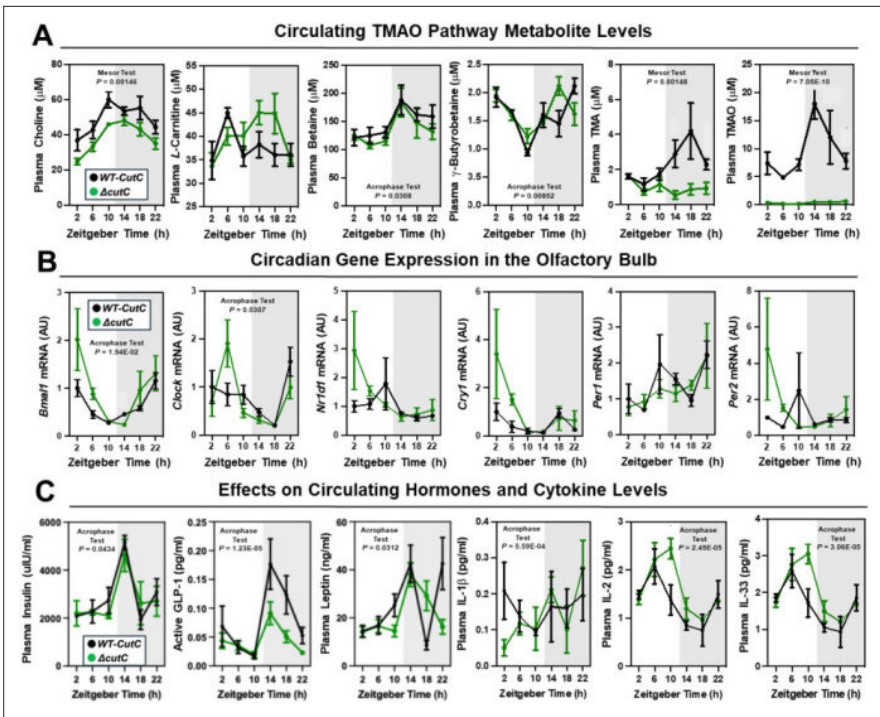

**Figure 4.** Transplanting a defined synthetic microbial community with or without genetically deleted trimethylamine production capacity (Δ*cutC*) alters host circadian rhythms. Germ-free C57Bl/6 mice (recipients) were gavaged with the core community (*B. caccae*, *B. ovatus*, *B. thetaiotaomicron*, *C. aerofaciens*, and *E. rectale*) with TMA producing wild-type (WT) *C. sporogenes* (produces TMAO) or *C. sporogenes* Δ*cutC*. Gnotobiotic mice were then necropsied at 4-hr intervals to collect tissues including plasma (**A, C**) and olfactory bulb (**B**). (**A**) Plasma levels of TMAO pathway metabolites (choline, L-carnitine, betaine, γ-butyrobetaine, trimethylamine (TMA), and trimethylamine *N*-oxide (TMAO)) were quantified by liquid chromatography–tandem mass spectrometry (LC–MS/MS). (**B**) PCR was performed on olfactory bulb to examine key circadian clock regulators. (**C**) Plasma levels of metabolic hormones (insulin, GLP-1, and leptin) and select cytokines including interleukins (IL-1β, IL-2, and IL-33) were measured as described in the Methods section. Data shown represent the means ± SEM for *n* = 5–6 individual mice per group. Differences between WT-*cutC* and Δ*cutC* groups were determined using cosinor analyses, and p-values are provided where there were statistically significant differences between groups for circadian statistics. The complete cosinor statistical analysis for circadian data can be found in **Supplementary file 1**. Significant differences between WT-*cutC* and Δ*cutC* groups were also analyzed by Student's *t*-tests within each ZT time point (*p < 0.05 and **p < 0.01).

The online version of this article includes the following figure supplement(s) for figure 4:

**Figure supplement 1.** The oscillatory patterns of the TMAO-defined community are altered when cutC is genetically deleted.

**Figure supplement 2.** Mice lacking either gut microbial TMA production or host-driven TMA oxidation have altered circadian rhythms.

---

to mice colonized with the control wild-type community (*Figure 4—figure supplement 2A*; *Supplementary file 1*). We next wanted to more comprehensively quantify a variety of other well-known metabolites that are known to originate from bacterial sources (*Figure 4—figure supplement 2B*). The rationale for performing microbe-focused metabolomics is that we have found if we alter one gut microbe-derived metabolite in other gnotobiotic mouse studies, there can be unexpected alterations in other distinct classes of microbe-derived metabolites (*Wang et al., 2023*). This is likely due to the fact that complex microbe–microbe and microbe–host interactions work together to define systemic levels of circulating metabolites, influencing both the production and turnover of distinct and unrelated metabolites. Although the targeted deletion of *cutC* in *C. sporogenes* prevents the production of TMA from choline, there were alterations of several other gut microbe-derived metabolites in mice colonized with the Δ*cutC C. sporogenes* community (*Figure 4—figure supplement 2*). Bacterially derived aromatic amino acid metabolites such as hippuric acid, indole-3-propionic acid, and indole

acetic acid were significantly reduced in mice colonized with the ΔcutC community (*Figure 4—figure supplement 2*). Also, the bacterially derived phenylalanine metabolites phenylacetic acid and phenylacetylglycine gained rhythmicity with marked increases in their peak amplitude in mice colonized with the ΔcutC community when compared to control mice (*Figure 4—figure supplement 2*).

Finally, we wanted to examine whether host co-metabolism of TMA can also alter circadian rhythms (*Figure 4—figure supplement 2*). Although gut microbes are the sole source of TMA, circulating levels are also shaped by abundant conversion of TMA to TMAO by the host liver enzyme FMO3 (*Schugar et al., 2017*; *Zhu et al., 2018*). To test whether the hepatic conversion of TMA to TMAO by FMO3 may alter circadian rhythms, we necropsied female *Fmo3$^{+/+}$* and *Fmo3$^{-/-}$* mice at ZT2 (early light cycle) or ZT14 (early dark cycle). *Fmo3$^{-/-}$* mice have reduced levels of TMAO at both ZT2 and ZT14, as well as reduced expression of key circadian genes *Bmal1* and *Per1* at ZT2 (*Figure 4—figure supplement 2*). Taken together, our data suggest that bacterial production, FMO3-driven metabolism of TMA, as well as sensing of TMA by the host receptor TAAR5, converge to shape circadian rhythms in gene expression, metabolic hormones, gut microbiome composition, and innate behaviors.

## Discussion

The metaorganismal TMAO pathway is strongly associated with many human diseases (*Wang et al., 2011*; *Tang et al., 2013*; *Koeth et al., 2013*; *Zhu et al., 2016*; *Skye et al., 2018*; *Bennett et al., 2013*; *Koeth et al., 2014*; *Leng et al., 2024*; *Schugar et al., 2017*; *Dehghan et al., 2020*; *Miao et al., 2015*; *Steinke et al., 2020*; *Tang et al., 2015*; *Zixin et al., 2022*; *Kumari et al., 2020*; *Vogt et al., 2018*; *Xu et al., 2015*; *Banerjee et al., 2024*; *Li et al., 2022*; *Heianza et al., 2017*; *Schiattarella et al., 2017*; *Jia et al., 2019*; *Zhou et al., 2023*; *Chen et al., 2019*; *Zhu et al., 2018*), which has prompted the rapid development of drugs intended to lower circulating levels of TMAO (*Wang et al., 2015*; *Roberts et al., 2018*; *Organ et al., 2020*; *Schugar et al., 2022*; *Helsley et al., 2022*; *Zhang et al., 2021*; *Gupta et al., 2020*; *Benson et al., 2023*). As the rapid drug discovery advances toward human studies, it will be extremely important to understand the diverse mechanisms by which the primary metabolite TMA and/or the secondary metabolite TMAO promotes disease pathogenesis in the human metaorganism. There is compelling evidence that the end product of the pathway TMAO can promote inflammation, ER stress, and platelet activation via activation of NFκB, the NLRP3 inflammasome, PERK, and stimulus-dependent calcium release, respectively (*Chen et al., 2019*; *Seldin et al., 2016*; *Sun et al., 2016*; *Chen et al., 2017*; *Zhang et al., 2020*). However, these TMAO-driven mechanisms only partially explain the links to so many diverse human diseases. Here, we provide new evidence that in addition to these TMAO-driven mechanisms, the primary gut microbe TMA can in parallel shape circadian rhythms through the host GPCR TAAR5. The major findings of the current studies are: (1) Mice lacking the TMA receptor TAAR5 have abnormal oscillations in core circadian genes, particularly in the olfactory bulb, (2) Compared to wild-type controls, *Taar5$^{-/-}$* mice have altered circulating levels of cytokines, metabolic hormones, and metabolites at certain times of the day, (3) *Taar5$^{-/-}$* mice have altered innate and repetitive behaviors that emerge only in a time of day-dependent manner, (4) The normal oscillatory behavior of the cecal microbiome is dysregulated in *Taar5$^{-/-}$* mice, (5) Genetic deletion of gut microbial choline TMA lyase activity, using defined *cutC*-null microbial communities in vivo, results in rewiring of circadian rhythms in gene expression, metabolic hormones, cytokines, and metabolites, (6) The bacterial strains in the defined microbial used in gnotobiotic mouse study oscillate differently depending on the presence of *cutC*, and (7) Mice lacking the host liver TMA-to-TMAO converting enzyme FMO3 likewise have altered circadian gene expression in the olfactory bulb (*Figure 5*). Collectively, our findings suggest that therapeutic strategies designed to limit gut microbial TMA production (i.e. TMA lyase inhibitors), host liver TMA oxidation (i.e. FMO3 inhibitors), or sensing of TMA by the host GPCR TAAR5 (i.e. TAAR5 inhibitors) will need to be carefully evaluated for pleiotropic effects on circadian rhythms. This body of work also demonstrates that diet–microbe–host interactions can powerfully shape chronobiology and provides one of the first examples of a microbial metabolite being sensed by a host GPCR to rewire circadian rhythms.

Currently, the only known host receptor that senses gut microbe-produced TMA is the volatile amine receptor TAAR5, which allows for species-specific recognition of the 'fishy' odor intrinsic to TMA but not TMAO (*Wallrabenstein et al., 2013*; *Li et al., 2013*; *Liberles, 2015*). In general, the trace amine-associated receptor (TAAR) subfamily of GPCRs primarily function as olfactory receptors in vertebrates (*Liberles, 2015*; *Freyberg and Saavedra, 2020*). Ligand-dependent activation

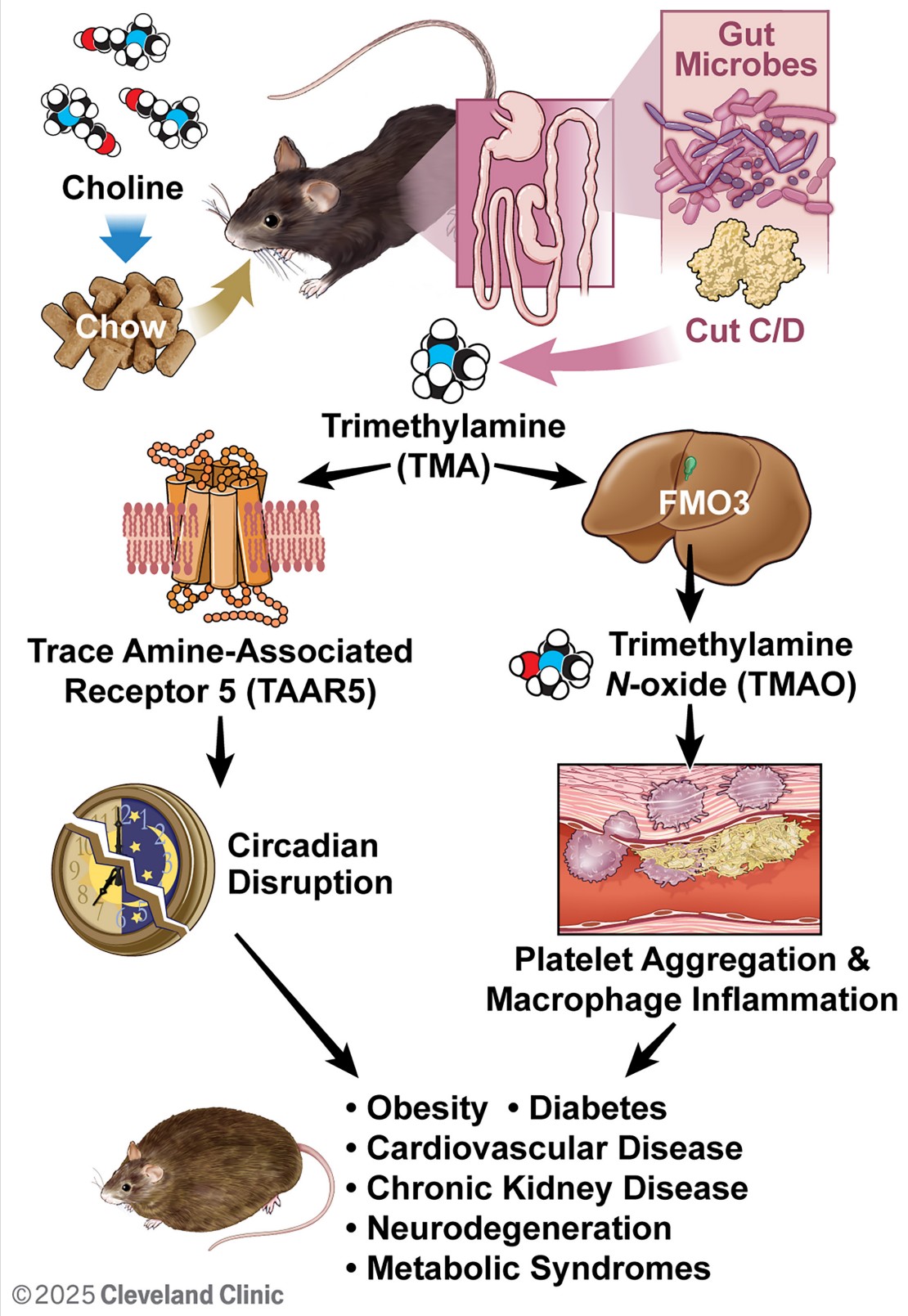

**Figure 5.** Summary of findings. Dietary choline is converted by gut microbial CutC/D into TMA, which signals through the host receptor TAAR5 or is converted to TMAO by hepatic FMO3. Loss of TAAR5 disrupts core circadian gene oscillations (particularly in the olfactory bulb), alters time-of-day regulation of cytokines, hormones, metabolites, and reveals time-dependent changes in innate and repetitive behaviors, alongside dysregulated oscillatory microbiome dynamics. Eliminating microbial cutC similarly rewires circadian oscillations in host immune and metabolic pathways, and

*Figure 5 continued on next page*

*Figure 5 continued*

microbial strains themselves exhibit altered rhythmicity depending on cutC status. Likewise, *Fmo3*[-/-] mice display disturbed circadian gene rhythms, together defining a microbial TMA–TAAR5–FMO3 axis as a key regulator of circadian control, inflammation, and metabolic disease-relevant physiology.

of each individual TAAR receptor occurs in a unique sensory neuron population primarily localized to olfactory cilia, and when activated typically elicits TAAR-specific olfactory-related behavior responses (*Liberles, 2015*; *Freyberg and Saavedra, 2020*). TMA-dependent activation of TAAR5 was first shown to promote attraction and social interaction between females and male mice in a very species and strain-specific manner (*Li et al., 2013*). This seminal study by Liberles and colleagues showed that the TMA–TAAR5 olfactory circuit can powerfully shape olfactory-driven social interaction, and FMO3-driven conversion of TMA to TMAO suppresses TAAR5 activation (*Li et al., 2013*). In parallel, we recently performed a comprehensive study to assess cognitive, motor, anxiolytic, social, olfactory, and innate behaviors in mice treated with small molecule choline TMA lyase inhibitors to block bacterial TMA production (*Massey et al., 2023*). Much like the olfactory and social phenotypes seen in *Taar5*[-/-] mice (*Li et al., 2013*), we found that pharmacologic blockade of bacterial TMA production significantly altered olfaction and olfactory-related social behaviors, but did not significantly alter cognitive, motor, and anxiolytic behaviors (*Massey et al., 2023*). It is interesting to note that choline TMA lyase inhibitors altered olfactory perception of several odorant stimuli beyond TMA itself including a cookie, almond, vanilla, corn oil, and coyote urine (*Massey et al., 2023*). This indicates that drugs blocking TMA–TAAR5 signaling could impact olfactory neurogenesis to impact sensing of diverse odorant cues that are well beyond the rotten fish smell intrinsic to TMA. Given that anosmia is a common occurrence in several TMAO-related diseases including CKD (*Morales Palomares et al., 2025*), obesity, diabetes (*Faour et al., 2022*), neurodegenerative diseases (*Chen and Kostka, 2024*; *Papazian and Pinto, 2021*), and COVID-19 (*Chang et al., 2024*), it is tempting to speculate that TMA-lowering therapeutics may hold promise to potentially restore olfactory perception to diverse stimuli in these disease conditions. However, additional studies are needed to formally test this hypothesis. Given the clear links between the TMA–TAAR5 signaling axis and olfactory-related behaviors in mice (*Massey et al., 2023*; *Li et al., 2013*), it will be important to carefully evaluate effects of TMAO-lowering drugs on olfaction in future human studies.

A commonality of many bacterially derived volatile metabolites is that they have a pungent odor which is sensed by olfactory neurons, and these microbe–host olfactory circuits shape many diverse behavioral and metabolic responses in the human host. For example, strong bacterially derived odors such as trimethylamine (i.e. smells like rotting fish), hydrogen sulfide (i.e. the rotten egg smell), polyamines including putrescine and cadaverine (i.e. the smell of rotting cadavers/roadkill) can serve as an effective 'don't eat me' signal for us to avoid consuming spoiled food. Within the human metaorganism, the gut microbiome and select gut microbe-derived metabolites play an important role in shaping not only food intake but also other aspects of host energy metabolism, insulin sensitivity, immunology, and the circadian clock (*Mukherji et al., 2013*; *Asher and Sassone-Corsi, 2015*; *Thaiss et al., 2014*; *Liang et al., 2015*; *Choi et al., 2021*). It is important to note that there is bi-directional communication between gut microbes and the host in maintaining the circadian rhythms in metabolism and immunity within mammalian metaorganisms (*Mukherji et al., 2013*; *Asher and Sassone-Corsi, 2015*; *Thaiss et al., 2014*; *Liang et al., 2015*; *Choi et al., 2021*). First, central and peripheral clock machinery in the host clearly shape circadian oscillations in the gut microbiome primarily through innate immune mechanisms (*Mukherji et al., 2013*; *Asher and Sassone-Corsi, 2015*). At the same time, circadian oscillations intrinsic to the gut microbiome itself can have profound impacts on host physiology and disease (*Thaiss et al., 2014*; *Liang et al., 2015*; *Choi et al., 2021*). Although there is extensive knowledge on how the host circadian clock can impact the oscillatory behavior of the gut microbiome (*Mukherji et al., 2013*; *Asher and Sassone-Corsi, 2015*), much less is known regarding mechanisms by which the gut microbiome can instruct the host circadian clock. Our data support the notion that gut microbe-derived TMA and TAAR5 activation can impact circadian rhythms in gene expression, metabolic hormones, gut microbiome composition, and innate behaviors. Using both pharmacologic (*Schugar et al., 2022*) and genetic approaches, we have demonstrated that blocking either gut microbial TMA production or sensing by the host TMA receptor TAAR5 results in striking rearrangement of circadian rhythms in the gut microbiome that are linked to metabolic homeostasis and olfactory-related behaviors in mice (*Schugar et al., 2022*; *Massey et al., 2023*). In further support

of the connections between the TMAO pathway and host circadian disruption, a recent paper showed that FMO3 inhibitors blocking host TMA oxidation improved learning and memory deficits driven by chronic sleep loss (*Zhu et al., 2024*). Other recent studies in humans also show that TMAO is elevated with sleep deprivation (*Giskeødegård et al., 2015*), and elevated TMAO levels are associated with an 'evening chronotype' that is linked to increased risk of cardiometabolic disease (*Barrea et al., 2021*). Collectively, the metaorganismal TMAO pathway plays an important role in integrating dietary cues and microbe–host interactions in circadian rhythms. Given TMAO-lowering drugs are rapidly advancing toward human studies, it will be of great interest to determine whether these drugs can improve circadian rhythms and potentially synergize with other developmental chronotherapies to improve human health.

## Limitations

First, it is critically important to point out that the TMA–TAAR5 signaling axis has evolved to elicit species-specific behavioral responses that differ quite substantially across rodent species and humans (*Li et al., 2013*; *Liberles, 2015*). When the 'stinky' fish odor of TMA is sensed by rat or human TAAR5, the result is typically aversion from consuming what is perceived as rotting food (*Li et al., 2013*; *Liberles, 2015*). However, select strains of mice have concentration-dependent attraction to low levels of TMA or aversion to high levels of TMA (*Li et al., 2013*). It has been theorized that this concentration-dependent behavioral response to TMA in mice may have evolved as a kairomone signal that allows mice to sense approaching prey animals (*Li et al., 2013*; *Liberles, 2015*). Clearly, the human TMA–TAAR5 signaling circuit evolved for different reasons, where predator–prey signaling circuits are less prominent. Therefore, additional studies are needed to better understand what diverse roles TMA-driven TAAR5 activation plays in the circadian rhythms in human metabolism and behavior. The recent discovery of humans with loss-of-function TAAR5 genetic variants (*Gisladottir et al., 2020*) may provide a unique population to confirm and extend the findings we report here in mice. Another limitation in our current understanding is that drugs blocking gut microbial TMA production (*Wang et al., 2015*; *Roberts et al., 2018*; *Organ et al., 2020*; *Schugar et al., 2022*; *Helsley et al., 2022*; *Zhang et al., 2021*; *Gupta et al., 2020*; *Benson et al., 2023*) or host TMA oxidation (*Schugar et al., 2017*; *Miao et al., 2015*; *Zhu et al., 2024*) have only been tested in mice. Although these drugs show clear benefit in mice models, the human gut microbiome is quite different from mice and the relative affinity of TMA for TAAR5 is ~200-fold less in humans when compared to rodent forms of the receptor (*Wallrabenstein et al., 2013*; *Li et al., 2013*; *Liberles, 2015*). Therefore, it is impossible to extrapolate findings in mice to humans. As TMAO-lowering drugs move into human trials, there is strong potential that human-specific phenotypes may emerge. Another limitation is that we have not rigorously studied diet-induced obesity, effects on food intake, or CVD/atherosclerosis in this study. Additional studies are needed to understand whether TMA-driven TAAR5 activation impacts cardiometabolic disease. A final limitation here is that here we have studied mice globally lacking TAAR5 in all tissues and cell types. Although TAAR5 has primarily been studied in olfactory neurons in the brain, TAAR5 is also expressed in immune cells (*Moiseenko et al., 2024*) as well as skeletal and cardiac muscle (*Schugar et al., 2022*). Given the expression of TAAR5 outside of the central nervous system, future studies utilizing cell-restricted *Taar5* knockout mice will provide clues into the cell autonomous roles for TMA–TAAR5 signaling. Despite these limitations, a growing body of evidence links TMA and TAAR5 activation to circadian regulation of metabolism and olfactory-related behaviors.

## Materials and methods
### Animal studies (mice)

*Taar5⁻/⁻* mice [Taar5^tm1.1(KOMP)Vlcg] used here were created from ES cell clone 10675A-A8 and originated on the C57BL/6N background by Regeneron Pharmaceuticals, Inc. The successfully targeted ES cells were then used to create live mice by the KOMP Repository and the Mouse Biology Program at the University of California Davis. It is important to note that this KO/reporter mouse line has the removal of the neomycin selection cassette and critical exon(s) leaving behind the inserted *lacZ* reporter sequence knocked into the *Taar5* locus. Given the well-appreciated differences in metabolic homeostasis between C57BL/6N and C57BL/6J substrains (*Fergusson et al., 2014*; *Attané et al., 2016*), we backcrossed this line >10 generations into the C57BL/6J congenic strain

and backcrossing was confirmed by mouse genome SNP scanning at the Jackson Laboratory (Bar Harbor, ME). Heterozygous mice on a pure C57BL/6J background were bred to generate littermate $Taar5^{+/+}$ and $Taar5^{-/-}$ mice. Global $Fmo3^{-/-}$ mice on a pure C57BL/6J background have been previously described (*Massey et al., 2022*). All mice were fed standardized control chow diet with defined sufficient level of dietary choline (Envigo diet TD:130104) that was twice irradiated for sterilization. For studies of circadian rhythms, 9-week-old C57Bl6/J male mice were adapted for 2 weeks with a strict 12:12 light:dark cycle. After 7 days, plasma and tissue were collected every 4 hr over a 24-hr period. All dark cycle necropsies were performed under red light conditions to avoid light-driven circadian cues. For the CutC mutant studies shown in main *Figure 4*, 7- to 8-week-old germ-free B6/N mice were purchased from Taconic and placed on a sterile chow. Two weeks later, the mice were given an oral gavage of our 6-member community that included *B. caccae*, *B. ovatus*, *B. theta*, *C. aerofaciens*, *E. rectale*, and either wild-type or mutant *C. sporogenes* suspended in sterile PBS + 20% glycerol as previously described (*Romano et al., 2017*). Gnotobiotic colonized mice were maintained using the Allentown Sentry SPP Cage System (Allentown, NJ) on a 14-hr:10-hr light:dark cycle. Mice were transferred to a room equipped with 12:12 light–dark cycling while still housed on Allentown Transport Carts. After 1 week of acclimation, plasma and tissue collection were performed every 4 hr over a 24-hr period. All dark cycle necropsies were performed under red light conditions. For all studies, plasma was collected by cardiac puncture, and liver, GWAT, BAT, and olfactory bulb were collected, flash-frozen, and stored at –80°C until the time of analysis. All mice were maintained in an Association for the Assessment and Accreditation of Laboratory Animal Care, International-approved animal facility, and all experimental protocols were approved by the Institutional Animal Care and Use Committee of the Cleveland Clinic (Approved IACUC protocol numbers 00001941, 00002609, and 0002866).

## RNA and realtime PCR methods

RNA isolation and quantitative polymerase chain reaction (qPCR) was conducted using methods previously described with minor modifications (*Schugar et al., 2017*; *Schugar et al., 2022*; *Massey et al., 2023*). To extract RNA, the Monarch Total RNA MiniPrep Kit (New England BioLabs, Inc) was employed, and 325 ng of cDNA was utilized for Realtime PCR on a 384-well plate, using Applied Biosystem's Power SYBR Green PC mix in Roche LightCycler 480 I1. The levels of induced mRNAs were standardized by the ΔΔCT method and were normalized to cyclophilin A. The specificity of the primers was verified by analyzing the melting curves of the PCR products. Primer information is available in the Star Methods table.

## Quantification of plasma metabolites using LC–MS/MS

Stable isotope dilution high-performance LC–MS/MS was used for quantification of levels of TMAO, TMA, choline, carnitine, and γ-butyrobetaine and all other reported circulating plasma metabolites as previously described performed on a Shimadzu 8050 triple quadrupole mass spectrometer (*Schugar et al., 2017*; *Schugar et al., 2022*; *Massey et al., 2023*). A subset of samples was also analyzed on a Thermo Vanquish liquid chromatograph coupled with a Thermo TSQ Quantiva mass spectrometer. 0.2% of formic acid in water (LC–MS grade) was used as mobile phase A and 0.2% formic acid in acetonitrile was used as mobile phase B. The separation was conducted using the following gradient: 0 min, 5% B; 0–2 min, 5% B; 2–8 min, 5–100% B; 8–16 min, 100% B; 16–16.5 min, 5% B; 16.5–25 min, 5% B. The flow rate was set at 0.2 ml/min. Samples were injected at 2 µl onto a Phenomenex Gemini C18 analytical column (2.0 × 150 mm, 3 µm). Column temperature was set at 25°C. The acquired data were processed by Thermo Xcalibur 4.3 software to calculate the concentrations. Both methods used multiple reaction monitoring of precursor and characteristic products as we have previously described (*Schugar et al., 2017*; *Schugar et al., 2022*; *Massey et al., 2023*).

## Plasma hormone and cytokine quantification

Plasma hormones and cytokine levels were quantified using U-PLEX (product # K15297K) and V-PLEX (K15297K) assays per the manufacturer's instructions (Meso Scale Diagnostics, Rockville, Maryland, USA).

## Olfactory cookie test

To broadly understand olfactory perception, we performed the olfactory cookie test using methods previously described (*Massey et al., 2023*). Briefly, 1 day prior to testing, food was removed from the home cage. The following day, a clean cage was filled with ~1.5 in. of clean bedding and a small portion of peanut butter cookie (Nutter Butter, Nabisco) was placed randomly ~1 cm under the bedding. A test mouse was placed into the cage and given 10 min to find the hidden cookie. Latency to find the cookie was recorded.

## Marble burying test

To test for innate/repetitive behavioral alterations, mice underwent a marble burying test. A clean mouse cage was filled with 2.5–3 in. of bedding. 20 black marbles were placed evenly in 5 rows of 4 across the bedding. A test mouse was placed into the cage and was allowed to bury marbles for 30 min. After 30 min, the number of marbles buried was counted. A marble was defined as buried when <25% of the marble was visible.

## Olfactory discrimination tests

These tests were performed as described previously (*Massey et al., 2023*). Briefly, a set of non-social odors (almond, banana, corn oil, and water) and two social odors were prepared. Three 6″ cotton-tipped applicators containing a single scent were prepared for each of the five odors. Each odor was presented three times in a row for 2 min. The cotton applicator was tapped to the wire cage lid on an empty clean home cage so that the cotton tip was hanging just below the wire lid. All trials were video recorded and the amount of time each mouse spent exploring the cotton tip was recorded.

## Three-box chamber tests

### Preference

Three-chamber social interaction test was performed using a three-chambered box as described previously (*Massey et al., 2023*). This test consisted of three 10 min trials. During the first trial, the mouse was allowed to explore the three-chamber box in which each end-chamber contained an empty cage (upside down pencil holder). This test was used to determine if mice had a clear preference for a chamber.

### Social

In the second trial, the three-chamber box contained a novel stimulus mouse under a cage in one of the end-chambers and an inanimate object (ping pong ball) under a cage in the opposite end-chamber. The test mouse was free to choose between a caged inanimate object and a caged, social target.

### Novel object

For the third trial, the test mouse was free to choose between a caged novel social target (novel mouse) versus the same caged mouse in trial 2 (familiar social target). Locations of inanimate targets and social targets were counterbalanced, and mice were placed back into the home cage for very brief intervals between trials. Total duration in chamber and interaction zone was recorded using video-tracking and automated Noldus EthoVision XT 14 (Leesburg VA).

## Social interaction with juvenile

Social interaction with a juvenile was performed in a novel empty home cage under dim white light as described previously (*Massey et al., 2023*). Briefly, following a 15-min habituation, a test mouse was placed in a novel empty cage with a juvenile stimulus mouse (BALB/cJ juvenile mouse; 3 weeks of age; Stock #000651 Jackson Labs) and allowed to directly interact for 2 min. Interaction was scored by observing the duration and number of times the test mouse-initiated contact with or sniffed the juvenile mouse. Contact was considered as any part of the boy touching the other mouse. Three days later, the same test mouse and juvenile mouse were paired again in a novel clean cage for 2 min and scored in the same manner.

## Startle test

Mice underwent a startle threshold test using methods previously described (*Massey et al., 2023*). Mice were placed inside holding cylinders that sit atop a piezoelectric accelerometer that detects and transduces animal movement (SR-LAB Startle Response System; SD Instruments; San Diego, CA). Acoustic stimuli were delivered by high-frequency speakers mounted 33 cm above the cylinders. Mouse movements were digitized and stored using computer software supplied by San Diego Instruments. For the startle reactivity test, mice were subjected to eight presentations of six trial types given in pseudorandom order: no stimulus, 80-, 90-, 100-, 110-, or 120-dB pulses. The mean startle amplitudes for each condition were calculated. Chambers were calibrated before each set of mice, and sound levels were monitored using a sound meter (Tandy).

## Forepaw grip strength test

Grip strength was measured using a digital grip meter (Chatillon Gauge with Mesh bar; Columbus Instruments, Columbus, OH) in which a mouse was suspended by its tail and allowed to grab onto a stainless-steel grid bar that is attached to a force transducer. The mouse was gently pulled away from the meter in a horizontal plane. The force applied to the bar immediately before the mouse released its grip was recorded in grams force by the digital sensor. Mice completed 10 trials with an intertrial interval of at least 30 s. The mouse's reported grip strength was the average of all 10 trials.

## Hotplate sensitivity test

For the hotplate sensitivity test, each mouse received a single trial that lasted a maximum of 30 s, and the behavioral response was scored live. To assess sensitivity to hot stimuli, a hot plate (Ugo Basile, Italy) was warmed to 52°C and a mouse was placed on the plate and a tall plexiglass cylinder was placed around the mouse to prevent the animal from leaving. A foot pedal was pressed to start the timer on the hotplate. As soon as the mouse showed a response to the stimuli such as jumping or licking their paws, the foot pedal was pressed again to stop the timer and the mouse was removed from the plate. Latency to respond was recorded.

## Rotarod

This was a 2-day task in which mice were placed onto a nonrotating rod (3 cm in diameter) that had ridges for a mouse to grip and was divided into four sections that allowed four mice to be tested at the same time (Rotamex-5; Columbus Instruments; Columbus, OH). Each mouse was subjected to 4 trials/day with an inter-trial interval of at least 30 min to prevent its performance from being impaired by fatigue. For each trial, a mouse was placed onto the rod and allowed to balance itself. When mice were balanced on the rod, the rod started rotating and accelerated from 4 to 40 rpm over a 5-min period. Latency to stay onto the rod was assessed with the use of an infrared beam just above the rod and was broken once the mouse fell off the rod. Each mouse was subjected to a total of 8 trials over a 2-day period (4 trials/day). Latency to fall (seconds) was recorded.

## Nesting

Nesting behavior was performed in a well-lit room by placing a mouse in a novel home cage with a cotton nestlet (5.5 × 5.5 × 0.5 cm) but no bedding. Height and width of the nests were measured at 30, 60, and 90 min.

## Fear conditioning

Fear conditioning (FC) consisted of a training period and a testing period. For the training period, mice were placed in the fear conditioning boxes (Med Associates; Fairfax VT), with a gridded floor, an inside white light and NIR light, and a fixed NIR camera for video recording. Mice were placed inside the chamber for 2 min, and then a 30-s, 90-dB acoustic conditioned stimulus (CS; white noise) co-terminated with a 2-s 0.6 mA foot shock (US). Mice received a total of three US–CS pairing, with each pairing separated by 1 min. The mouse remained in the chamber for 30 s after the pairings before returning to its home cage. The testing period occurred 24 hr after the training period ended. The mouse's freezing behavior (motionless except for respirations) was monitored during testing to assess memory. Prior to cue FC testing, the inside of the FC chamber was modified using white plastic sheets, interior lighting, and vanilla extract on a paper towel placed in the bedding chamber to make

the chamber look and smell differently than the training period. Mice were placed into this modified chamber and allowed to habituate for 3 min. Following 3 min the same white noise tone used in training was presented to the mouse for 3 min. During the tone presentation, mouse freezing behavior was recorded. Total percent freezing for cue FC was recorded.

### Elevated plus maze

The elevated plus maze task was conducted essentially as described previously (*Massey et al., 2023*). Mice were placed in the center of a black, plexiglass elevated plus maze (each arm 33 cm long and 5 cm wide with 25 cm high walls on closed arms) in a dimly lit room for 5 min. Automated video tracking software from Noldus Ethovision XT13 (Leesburg, VA) was used to track time spent in the open and closed arms, number of open and closed arm entries, and number of explorations of the open arm (defined as placing head and two limbs into open arm without full entry).

### Y-maze

Spontaneous alteration in Y-maze was assessed as previously described (*Massey et al., 2023*). Briefly, a mouse was placed into one arm of the Y-maze facing the center (14″). Video tracking was used to record the spontaneous behavior of each mouse for 10 min. Zone (i.e. arm) alteration was later analyzed using Noldus EthoVision XT14, to determine when a mouse entered three consecutive different arms of the maze. Spontaneous alteration % was calculated as follows: Alteration % = #spontaneous alterations/total number of arm entries – 2 × 100.

### Open field

The open field was conducted as described previously (*Massey et al., 2023*). Briefly, mice were placed along the edge of an open arena (44 × 44 × 44 cm) and allowed to freely explore for 10 min. Time spent in the center of the arena 15 × 15 cm as well as locomotor activity was measured. Mice were monitored using Noldus EthoVision XT14 (Leesburg, VA).

### Morris water maze

A 48″ diameter, white, plastic, circular, heated pool was filled to the depth of 23.5″ with 22 + 1°C water made opaque with gothic white, nontoxic, liquid tempera paint in a room with prominent extra-maze cues. Mice were placed in one of the four starting locations facing the pool wall and allowed to swim until finding a 15-cm diameter clear platform for 20 s before being removed to the home cage. If mice did not find the platform within 60 s, they were guided to the platform by the experimenter and remained on the platform for 20 s before being removed to the home cage. Latency to reach the platform, distance traveled to reach the platform, swim speed, and time spent in each of four quadrants were obtained using automated video tracking software from Noldus (Ethovision XT13, Leesburg, VA). Mice were trained with 4 trials/day with an intertrial interval of 1–1.5 min for 10 consecutive days between 10 am and 3 pm. A probe trial (free swim with the submerged platform removed) was performed as the first trial of the day on days 11. Percent time spent in the target quadrant was calculated. Latency to platform, distance to platform, swim speed, and time spent in the target quadrant was analyzed.

### Indirect calorimetry and metabolic cage measurements during cold challenge

Cold-induced metabolic responses were quantified using the Oxymax CLAMS system (Columbus Instruments) as previously described (*Schugar et al., 2017*). Briefly, weight-matched *Taar5*$^{-/-}$ and *Taar5*$^{+/+}$ mice were acclimated to metabolic cages for 48 hr. Thereafter, physical activity, oxygen consumption (VO$_2$), carbon dioxide production (VCO$_2$), and respiratory exchange ratio (RER) were continually monitored for 24 hr at thermoneutrality (30°C), 24 hr at room temperature (22°C), and 24 hr in the cold (4°C). Data represent the last 6 a.m. to 6 a.m. period after adequate acclimation.

### 16S rRNA gene amplicon sequencing and bioinformatics

16S rRNA gene amplicon sequencing and bioinformatics analysis were performed using our published methods using mouse cecum samples (*Schugar et al., 2022*). Briefly, raw 16S amplicon sequence and metadata were *demultiplexed using split_libraries_fastq*.py script implemented *in QIIME2* (*Bolyen*

*et al., 2019*). Demultiplexed fastq file was split into sample-specific fastq files using the split_sequence_file_on_sample_ids.py script from QIIME2. Individual fastq files without non-biological nucleotides were processed using the Divisive Amplicon Denoising Algorithm (DADA) pipeline (*Callahan et al., 2016*). The output of the dada2 pipeline [feature table of amplicon sequence variants (an ASV table)] was processed for alpha and beta diversity analysis using *phyloseq* (*McMurdie and Holmes, 2013*), and microbiomeSeq (https://github.com/umerijaz/microbiomeSeq, *Ijaz, 2019*) packages in R. We analyzed variance (ANOVA) among sample categories while measuring the $\alpha$-diversity measures using plot_anova_diversity function in *microbiomeSeq* package. Permutational multivariate analysis of variance (PERMANOVA) with 999 permutations was performed on all principal coordinates obtained during CCA with the *ordination* function of the *microbiomeSeq* package. Pairwise correlation was performed between the microbiome (genera) and metabolomics (metabolites) data was performed using the microbiomeSeq package.

## 16S statistical analysis

Differential abundance analysis was performed using the random-forest algorithm, implemented in the DAtest package. Briefly, differentially abundant methods were compared with false discovery rate, area under the (receiver operator) curve, empirical power (Power), and false positive rate. Based on the DAtest's benchmarking, we selected lefseq and anova as the methods of choice to perform differential abundance analysis. We assessed the statistical significance ($p < 0.05$) throughout, and whenever necessary, we adjusted p-values for multiple comparisons according to the Benjamini and Hochberg method to control False Discovery Rate (*Benjamini and Hochberg, 1995*). Linear regression (parametric test) and Wilcoxon (non-parametric) test were performed on genera and ASV abundances against metadata variables using their base functions in R (version 4.1.2;).

## Metagenomic sequencing to detect oscillatory patterns occurring in gnotobiotic mice colonized with defined synthetic communities with or without CutC

Microbial community DNA extraction and sequencing was performed using methods published by our group earlier (*Lundy et al., 2021*). Briefly, DNA extraction from the cecum was performed using QIAGEN's DNeasy PowerSoil Pro kit. To assess sample integrity, an e-gel (electronic agarose gel) was run. The DNA concentration of each sample was then quantified using a Qubit Fluorometer, which measured the values in ng/µl. Using this concentration data, we prepared each sample for Illumina's Nextera XT library preparation kit, targeting a fragment size of 300–500 bp. After library construction, the fragment size distribution of each library was checked using Agilent's TapeStation 4200. The libraries were then re-quantified with the Qubit Fluorometer. Next, the libraries were pooled in a manner that ensured an even distribution of sequencing reads across all samples. The pooled library was quantified again using both a Qubit Fluorometer and qPCR to determine its molarity in nM. Finally, the pooled library was loaded onto the sequencer for sequencing.

## Data analyses for circadian rhythmicity (cosinor analyses)

A single cosinor analysis was performed as previously described (*Cornelissen, 2014*; *Fernandes et al., 2014*) Briefly, a cosinor analysis was performed on each sample using the equation for cosinor fit as follows:

$$Y(t) = M + A\cos(2\theta/\tau + \phi),$$

where *M* is the MESOR (midline statistic of rhythm, a rhythm adjusted mean), *A* is the amplitude (a measure of half the extent of the variation within the cycle), *Φ* is the acrophase (a measure of the time of overall highest value), and *τ* is the period. The fit of the model was determined by the residuals of the fitted wave. After a single cosinor fit for all samples, linearized parameters were then averaged across all samples allowing for calculation of delinearized parameters for the population mean. A 24-hr period was used for all analysis. Comparison of population MESOR, amplitude, and acrophase was performed as previously described (*Schugar et al., 2022*). Comparisons are based on *F*-ratios with degrees of freedom representing the number of populations and total number of subjects. All analyses were done in R v.4.0.2 using the cosinor and cosinor2 packages (*Fernandes et al., 2014*; *Nelson et al., 1979*; *Bingham et al., 1982*).

## Standard statistical analyses

Data are expressed as the mean ± SEM. All data were analyzed using either one- or two-way analysis of variance followed by Student's $t$-tests for post hoc analysis. Differences were considered significant at $p < 0.05$. All analyses were performed using GraphPad Prism software (version 10.2.2; GraphPad Software, Inc).

## Materials availability

All the data and materials that support the findings of this study are available within the article. Additional details can be acquired via a direct request to the corresponding author.

## Acknowledgements

This work was supported in part by National Institutes of Health grants R01 DK130227 (JMB), P01 HL147823 (JMB), P50 AA024333 (JMB), RF1 NS133812 (JMB), and an American Heart Association Postdoctoral Fellowship 24POST1178494 (SD). We are grateful to Dr. Federico Rey (University of Wisconsin – Madison, USA) for providing bacterial strains used in the mouse gnotobiotic studies performed here. We are also grateful to Dr. Stan Hazen for providing access to mass spectrometer instruments dedicated to targeted metabolomic methods necessary to quantify diverse gut microbe-derived metabolites.

## Additional information

### Competing interests

Zeneng Wang: Z.W. reports being named as co-inventor on pending and issued patents (#US-12055535-B2, #US-11835503-B2, #US-20220065829-A1) held by the Cleveland Clinic relating to cardiovascular diagnostics and therapeutics. Z.W. also reports having received royalty payments for inventions or discoveries related to cardiovascular diagnostics or therapeutics from Cleveland Heart Lab, a fully owned subsidiary of Quest Diagnostics and Procter & Gamble. Jonathan Mark Brown: J.M.B. reports being named as co-inventor on pending and issued patents (#US-20200121615-A1) held by the Cleveland Clinic relating to choline trimethylamine lyase inhibitors as therapies for cardiometabolic disease including obesity and type 2 diabetes. The other authors declare that no competing interests exist.

### Funding

| Funder | Grant reference number | Author |
| --- | --- | --- |
| National Institute of Diabetes and Digestive and Kidney Diseases | R01 DK130227 | Jonathan Mark Brown |
| National Heart Lung and Blood Institute | P01 HL147823 | Jonathan Mark Brown |
| National Institute on Alcohol Abuse and Alcoholism | P50 AA024333 | Jonathan Mark Brown |
| National Institute of Neurological Disorders and Stroke | RF1 NS133812 | Jonathan Mark Brown |
| American Heart Association | 10.58275/aha.24post1178494.pc.gr.190863 | Sumita Dutta |

The funders had no role in study design, data collection, and interpretation, or the decision to submit the work for publication.

### Author contributions

Kala K Mahen, Conceptualization, Formal analysis, Validation, Investigation, Methodology, Writing – original draft, Writing – review and editing; William J Massey, Naseer Sangwan, Zeneng Wang,

Formal analysis, Validation, Investigation, Methodology, Writing – review and editing; Danny Orabi, Amanda L Brown, Thomas C Jaramillo, Nour Mouannes, Venkateshwari Varadharajan, Xiayan Ye, Dante M Yarbrough, Natalie Zajczenko, Rachel Hohe, Dev Laungani, Adeline M Hajjar, Investigation, Methodology, Writing – review and editing; Amy Burrows, Anthony J Horak, Sumita Dutta, Marko Mrdjen, Lucas J Osborn, Rakhee Banerjee, Pranavi Linga, Mohammed Dwidar, Jennifer A Buffa, Validation, Investigation, Methodology, Writing – review and editing; Treg Grubb, Formal analysis; Garth R Swanson, Formal analysis, Methodology, Writing – review and editing; Jonathan Mark Brown, Conceptualization, Formal analysis, Supervision, Funding acquisition, Writing – original draft

### Author ORCIDs
Anthony J Horak ⓘ https://orcid.org/0009-0009-2024-3679
Mohammed Dwidar ⓘ https://orcid.org/0000-0003-1366-0393
Zeneng Wang ⓘ https://orcid.org/0000-0002-6455-8228
Jonathan Mark Brown ⓘ https://orcid.org/0000-0003-2708-7487

### Ethics
All mice were maintained in an Association for the Assessment and Accreditation of Laboratory Animal Care, International-approved animal facility, and all experimental protocols were approved by the Institutional Animal Care and Use Committee of the Cleveland Clinic (Approved IACUC protocol numbers 00001941, 00002609, and 0002866).

Reviewer #1 (Public review): https://doi.org/10.7554/eLife.107037.3.sa1
Reviewer #2 (Public review): https://doi.org/10.7554/eLife.107037.3.sa2
Reviewer #3 (Public review): https://doi.org/10.7554/eLife.107037.3.sa3
Author response https://doi.org/10.7554/eLife.107037.3.sa4

---

## Additional files

### Supplementary files
Supplementary file 1. Cosinor analyses for the entire manuscript.
MDAR checklist

### Data availability
The 16S microbiome data from wild-type and Taar5 knockout mice are publicly available at the following link: https://doi.org/10.5281/zenodo.15802940. Code used for cosinor analyses: cosinor: Tools for estimating and predicting the cosinor model, version 1.1. R package: https://CRAN.R-project.org/package=cosinor. Any additional information required to reanalyze the data reported in this paper is available from the lead contact (Dr. J. Mark Brown) upon request.

The following dataset was generated:

| Author(s) | Year | Dataset title | Dataset URL | Database and Identifier |
|---|---|---|---|---|
| Brown M, Sangwan N | 2025 | Gut Microbe-Derived Trimethylamine Shapes Circadian Rhythms Through the Host Receptor TAAR5 | https://doi.org/10.5281/zenodo.15802940 | Zenodo, 10.5281/zenodo.15802940 |

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

# Appendix 1

## Appendix 1—key resources table

| Reagent type (species) or resource | Designation | Source or reference | Identifiers | Additional information |
|---|---|---|---|---|
| Strain, strain background | *Bacteroides thetaiotaomicron* | ATCC | VPI-5482 | |
| Strain, strain background | *Bacteroides caccae* | ATCC | ATCC 43185 | |
| Strain, strain background | *Bacteroides ovatus* | ATCC | ATCC 8483 | |
| Strain, strain background | *Collinsella aerofaciens* | ATCC | ATCC 25986 | |
| Strain, strain background | *Eubacterium rectale* | ATCC | ATCC 33656 | |
| Strain, strain background | *Clostridium sporogenes* | ATCC | ATCC 15579 | |
| Strain, strain background | *Clostridium sporogenes ΔcutC* | Dr. Michael Fischbach at Stanford University | N/A | Mutant strain |
| Chemical compound, drug | Choline chloride | Sigma | #C7017 | LC/MS standard |
| Chemical compound, drug | Choline chloride (trimethyl-D$_9$, 98%) | Cambridge Isotope Laboratories, Inc | #DLM-549-MPT-PK | LC/MS standard |
| Chemical compound, drug | Betaine | Sigma | #61962 | LC/MS standard |
| Chemical compound, drug | *N*-(Carboxymethyl)-*N,N,N*-trimethyl-d9-ammonium chloride | C/D/N Isotopes Inc | #D-3352 | LC/MS standard |
| Chemical compound, drug | L-Carnitine hydrochloride | Sigma | #94954 | LC/MS standard |
| Chemical compound, drug | L-Carnitine-d3 HCl (*N*-methyl-d3) | C/D/N Isotopes Inc | #D-5069 | LC/MS standard |
| Chemical compound, drug | (3-Carboxypropyl)trimethylammonium chloride (butyrobetaine) | Sigma | #403245 | LC/MS standard |
| Chemical compound, drug | Butyrobetaine-d9 | **Wang et al., 2015** | NA | LC/MS standard |
| Chemical compound, drug | 5-Hydroxyindole-3-acetic acid | Sigma | #H8876 | LC/MS standard |
| Chemical compound, drug | 5-Hydroxyindole-3-acetic-2,2-D2 Acid | CDN Isotopes | #D-1547 | LC/MS standard |
| Chemical compound, drug | Hippuric acid | Sigma | #8206490100 | LC/MS standard |
| Chemical compound, drug | *N*-Benzoyl-d5-glycine | CDN Isotopes | #D-5588 | LC/MS standard |
| Chemical compound, drug | 4-OH-Hippuric acid | Santa Cruz Biotechnology | #SC-277427 | LC/MS standard |
| Chemical compound, drug | 3-Hydroxyhippuric acid | TRC | #H943125 | LC/MS standard |
| Chemical compound, drug | 2-Hydroxyhippuric acid | Carbosynth | #FH240191601 | LC/MS standard |
| Chemical compound, drug | Indole-3-propionic acid | Alfa Aesar | #L04877 | LC/MS standard |
| Chemical compound, drug | Indole-3-propionic-2,2-D2 acid | CDN Isotopes | #D-7686 | LC/MS standard |
| Chemical compound, drug | Methylindole-3-acetate | Sigma | #I9770 | LC/MS standard |
| Chemical compound, drug | DL-Indole-3-lactic acid | Chem-Impex | #21729 | LC/MS standard |
| Chemical compound, drug | *Phenylacetic acid* | Aldrich | #P16621 | LC/MS standard |

*Appendix 1 Continued on next page*

*Appendix 1 Continued*

| Reagent type (species) or resource | Designation | Source or reference | Identifiers | Additional information |
|---|---|---|---|---|
| Chemical compound, drug | *Phenylacetic Acid-1,2-13C2* | Chem Cruz | #SC-236371 | LC/MS standard |
| Chemical compound, drug | *DL-p-Hydrophenyllactic acid* | Aldrich | #H3253 | LC/MS standard |
| Chemical compound, drug | *Indoxyl-glucuronide* | Abcam | #Ab146380 | LC/MS standard |
| Chemical compound, drug | *L-Tryptophan-2',4',5',6',7'-d5 (indole-d5)* | CDN Isotopes | #D-1522 | LC/MS standard |
| Chemical compound, drug | *Serotonin* | Sigma | #H-9523 | LC/MS standard |
| Chemical compound, drug | *Tryptamine* | Aldrich | #193747 | LC/MS standard |
| Chemical compound, drug | *Tryptamine-α,α,β,β-D4 HCl* | CDN Isotopes | #D-1546 | LC/MS standard |
| Chemical compound, drug | *3-Indoleacetic acid* | Aldrich | #I3750 | LC/MS standard |
| Chemical compound, drug | *Indole-3-acetic-2,2-D2 acid* | CDN Isotopes | #D-1709 | LC/MS standard |
| Chemical compound, drug | Sodium phenyl sulfate | Enamine | #EN300-1704008 | LC/MS standard |
| Chemical compound, drug | TMAO | Sigma | #317594 | LC/MS standard |
| Chemical compound, drug | Trimethylamine $N$-oxide ($D_9$, 98%) | Cambridge Isotope Laboratories, Inc | #DLM-4779-1 | LC/MS standard |
| Chemical compound, drug | Phenylacetyl-L-glutamine | Chem-Impex | #16414 | LC/MS standard |
| Chemical compound, drug | N-alpha-(phenyl-d5-acetyl)-L-glutamine | CDN Isotopes | #D-6900 | LC/MS standard |
| Chemical compound, drug | Potassium p-tolyl sulfate P-cresol sulfate K-salt | TCI | #P2091 | LC/MS standard |
| Chemical compound, drug | p-Cresol sulfate, potassium salt ($D_7$, 98%) | Cambridge Isotope Laboratories, Inc | #DLM-9786 | LC/MS standard |
| Chemical compound, drug | 3-Indoxyl sulfate potassium salt | Chem-Impex | #21710 | LC/MS standard |
| Chemical compound, drug | 3-Indoxyl sulfate-d4 potassium salt | TRC | #I655102 | LC/MS standard |
| Chemical compound, drug | Phenylacetylglycine | Bachem | #4016439 | LC/MS standard |
| Chemical compound, drug | trans-3-Indoleacrylic acid | Aldrich | #I3807 | LC/MS standard |
| Chemical compound, drug | 4-Ethylphenyl sulfate potassium salt | TRC | #E925865 | LC/MS standard |
| Chemical compound, drug | Trimethylamine hydrochloride | Sigma | #41284 | LC/MS standard |
| Chemical compound, drug | Trimethylamine:DCl ($D_{10}$, 98%) | Cambridge Isotope Laboratories, Inc | #DLM-1817-5 | LC/MS standard |
| Commercial assay, kit | Monarch Total RNA Miniprep Kit | New England BioLabs | #T2010S | |
| Commercial assay, kit | V-Plex Cytokine Panel 1 Mouse Kit | Mesoscale Discovery | #K15245D | |
| Commercial assay, kit | V-Plex Pro-Inflammatory Panel 1 Mouse Kit | Mesoscale Discovery | #K15048D | |
| Commercial assay, kit | U-Plex Metabolic Combo Mouse Kit | Mesoscale Discovery | #K15297K | |
| Commercial assay, kit | Ultra Sensitive Mouse Insulin ELISA | Crystal Chem Inc | #90080 | |
| Commercial assay, kit | DNeasy PowerSoil Pro Kit | QIAGEN | #47014 | |
| Strain, strain background | Germ/free C57BL/6NTac | Taconic | Stock #: B6-GF-F | |

*Appendix 1 Continued on next page*

*Appendix 1 Continued*

| Reagent type (species) or resource | Designation | Source or reference | Identifiers | Additional information |
|---|---|---|---|---|
| Strain, strain background | C57BL/6J | Jackson | #00664 | |
| Strain, strain background | C57BL/6J *Taar5*<sup>-/-</sup> | This paper | NA | Created from ES cell clone 10675A-A8 and originated on the C57BL/6N background by Regeneron Pharmaceuticals, Inc |
| Strain, strain background | C57BL/6J *Fmo3*<sup>-/-</sup> | Massey et al. (**Attané et al., 2016**) | NA | |
| Sequence-based reagent | *LacZ* Forward | Sigma | PCR primer | CCAACGTGACCTATCCCATTAC |
| Sequence-based reagent | *LacZ* Reverse | Sigma | PCR primer | ATCTTCCTGAGGCCGATACT |
| Sequence-based reagent | *Bmal1* Forward | Sigma | PCR primer | CCAAGAAAGTATGGACACAGACAAA |
| Sequence-based reagent | *Bmal1* Reverse | Sigma | PCR primer | GCATTCTTGATCCTTCCTTGGT |
| Sequence-based reagent | *Nr1d1* Forward | Sigma | PCR primer | ATGCCAATCATGCATCAGGT |
| Sequence-based reagent | *Nr1d1* Reverse | Sigma | PCR primer | CCCATTGCTGTTAGGTTGGT |
| Sequence-based reagent | *Per1* Forward | Sigma | PCR primer | TGTCCTGGTTTCGAAGTGTG |
| Sequence-based reagent | *Per1* Reverse | Sigma | PCR primer | TGTGTCAAGCAGGTTCAGG |
| Sequence-based reagent | *Per2* Forward | Sigma | PCR primer | GCTGACGCACACAAAGAACT |
| Sequence-based reagent | *Per2* Reverse | Sigma | PCR primer | TAGCCTTCACCTGCTTCACG |
| Sequence-based reagent | *Clock* Forward | Sigma | PCR primer | AGGCACAGACATTATCGG |
| Sequence-based reagent | *Clock* Reverse | Sigma | PCR primer | ACCGTCTCATCAAGGGAC |
| Sequence-based reagent | *Cry1* Forward | Sigma | PCR primer | TACTGGGAAACGCTGAACCC |
| Sequence-based reagent | *Cry1* Reverse | Sigma | PCR primer | ACCCCAAGCTTGTTGCCTAA |
| Sequence-based reagent | *Cry2* Forward | Sigma | PCR primer | GCTGGAAGCAGCCGAGGAACC |
| Sequence-based reagent | *Cry2* Reverse | Sigma | PCR primer | GGGCTTTGCTCACGGAGCGA |
| Sequence-based reagent | *Prdm16* Forward | Sigma | PCR primer | CAGCACGGTGAAGCCATTC |
| Sequence-based reagent | *Prdm16* Reverse | Sigma | PCR primer | GCGTGCATCCGCTTGTG |
| Sequence-based reagent | *Ucp1* Forward | Sigma | PCR primer | ACTGCCACACCTCCAGTCATT |
| Sequence-based reagent | *Ucp1* Reverse | Sigma | PCR primer | CTTTGCCTCACTCAGGATTGG |
| Sequence-based reagent | *Pemt* Forward | Sigma | PCR primer | TGTGCTGTCCAGCTTCTATG |
| Sequence-based reagent | *Pemt* Reverse | Sigma | PCR primer | GAAGGGAAATGTGGTCACTCT |
| Sequence-based reagent | *Pdk4* Forward | Sigma | PCR primer | GTGCTCTCTGGTCCTCTGTG |
| Sequence-based reagent | *Pdk4* Reverse | Sigma | PCR primer | AGTCCAACGGACAAAACGGA |
| Sequence-based reagent | *Fmo3* Forward | Sigma | PCR primer | CCCACATGCTTTGAGAGGAG |

*Appendix 1 Continued on next page*

*Appendix 1 Continued*

| Reagent type (species) or resource | Designation | Source or reference | Identifiers | Additional information |
|---|---|---|---|---|
| Sequence-based reagent | *Fmo3* Reverse | Sigma | PCR primer | GGAAGAGTTGGTGAAGACCG |
| Sequence-based reagent | *CycloA* Forward | Sigma | PCR primer | GCGGCAGGTCCATCTACG |
| Sequence-based reagent | *CycloA* Reverse | Sigma | PCR primer | GCCATCCAGCCATTCAGTC |
| Sequence-based reagent | *Gapdh* Forward | Sigma | PCR primer | CCTCGTCCCGTAGACAAAATG |
| Sequence-based reagent | *Gapdh* Reverse | Sigma | PCR primer | TGAAGGGGTCGTTGATGGC |
| Software, algorithm | GraphPad Prism | https://www.graphpad.com/ | Version 10.4.1 | Statistical analysis/figures |
| Software, algorithm | DADA2 | https://benjjneb.github.io/dada2/ | Version 3.16 | Cosinor analysis |
| Software, algorithm | metagenomeSeq | https://github.com/HCBravoLab/metagenomeSeq; **Paulson, 2024** | Version: Release (3.20) | Cosinor analysis |
| Software, algorithm | DAtest | https://github.com/Russel88/DAtest; **Russel, 2022** | Version 2.8.0 | Cosinor analysis |
| Software, algorithm | DAtest | https://github.com/Russel88/DAtest | Version 2.8.0 | Cosinor analysis |
| Software, algorithm | ggplot2 | https://cran.r-project.org/web/packages/ggplot2/index.html | Version 3.5.1 | Cosinor analysis |
| Software, algorithm | cosinor | https://cran.r-project.org/web/packages/cosinor/index.html | Version 1.2.3 | Cosinor analysis |
| Software, algorithm | cosinor2 | https://cran.r-project.org/web/packages/cosinor2/index.html | Version 0.2.1 | Cosinor analysis |
| Software, algorithm | EthoVision XT15 (video tracking software) | Noldus | NA | Behavioral testing |
| Other | qScript | QuantaBio | #95048-100 | Realtime PCR |
| Other | Fast SYBR Green Master Mix | Applied Biosystems | #4385612 | Realtime PCR |
| Other | Nutter Butter Cookies | Nabisco | NA | Behavioral testing |
| Other | Marbles | Moon Marble Company | NA | Behavioral testing |
| Other | SR-LAB-Startle Response System | San Diego Instruments | #2325-0400 | Behavioral testing |
| Other | Elevated Plus Maze | Nationwide Plastics | NA | Behavioral testing |
| Other | Y Maze | Nationwide Plastics | NA | Behavioral testing |
| Other | Forepaw Grip Strength Meter | Columbus Instruments | #1027SM | Behavioral testing |
| Other | Rotamex-5 | Columbus Instruments | #0254-8000 | Behavioral testing |
| Other | Fear Conditioning Chambers | Med Associates | #VFC-008 | Behavioral testing |
| Other | Hot Plate | Ugo Basile | #35300 | Behavioral testing |
| Other | OxyMax Clams Home Cage System | Columbus Instruments | NA | Behavioral testing |

