## [Editor Report · eLife Assessment]

This study presents an **important** finding linking the bacterial metabolite trimethylamine and its receptor to circadian rhythms and olfaction. The current evidence supporting the claims of the authors is **compelling**. This work will be of broad interest to researchers interested in nutrition, microbial metabolism, circadian rhythms, and host-microbiome interactions.

---

## [Referee Report · Reviewer #1 (Public review)]

Summary:

This study focuses on the bacterial metabolite TMA, generated from dietary choline. These authors and others have previously generated foundational knowledge about the TMA metabolite TMAO, and its role in metabolic disease. This study extends those findings to test whether TMAO's precursor, TMA, and its receptor TAAR5 are also involved and necessary for some of these metabolic phenotypes. They find that mice lacking the host TMA receptor (Taar5-/-) have altered circadian rhythms in gene expression, metabolic hormones, gut microbiome composition, and olfactory and innate behavior. In parallel, mice lacking bacterial TMA production or host TMA oxidation have altered circadian rhythms.

Strengths:

These authors use state-of-the-art bacterial and murine genetics to dissect the roles of TMA, TMAO, and their receptor in various metabolic outcomes (primarily measuring plasma and tissue cytokine/gene expression). They also follow a unique and unexpected behavioral/olfactory phenotype. Statistics are impeccable.

---

## [Referee Report · Reviewer #2 (Public review)]

Summary:

In the manuscript by Mahen et al., entitled "Gut Microbe-Derived Trimethylamine Shapes Circadian Rhythms Through the Host Receptor TAAR5," the authors investigate the interplay between a host G protein-coupled receptor (TAAR5), the gut microbiota-derived metabolite trimethylamine (TMA), and the host circadian system. Using a combination of genetically engineered mouse and bacterial models, the study demonstrates a link between microbial signaling and circadian regulation, particularly through effects observed in the olfactory system. Overall, this manuscript presents a novel and valuable contribution to our understanding of host-microbe interactions and circadian biology. The addition of new data following revision adds mechanistic depth to more fully support the authors' conclusions.

Strengths:

(1) The manuscript addresses an important and timely topic in host-microbe communication and circadian biology.

(2) The studies employ multiple complementary models, e.g., Taar5 knockout mice, microbial mutants, which enhances the depth of the investigation.

(3) The integration of behavioral, hormonal, microbial, and transcript-level data provides a multifaceted view of the observed phenotype.

(4) Inclusion of rhythmic analysis of a defined microbial community adds novelty and strength to the overall findings.

(5) The identification of olfactory-linked circadian changes in the context of gut microbes adds a novel perspective to the field.

Weaknesses:

(1) While the authors suggest a causal role for TAAR5 and its ligand in circadian regulation, some of the data remain correlative in this context; however, the authors have appropriately tempered these claims, and mechanistic experiments are proposed to expand upon their compelling findings in future work.

---

## [Referee Report · Reviewer #3 (Public review)]

Summary:

Deletion of the TMA-sensor TAAR5 results in circadian alterations in the gene expression, particularly in the olfactory bulb; plasma hormones; and neurobehaviors.

Strengths:

Genetic background was rigorously controlled.

Comprehensive characterization.

Impact:

These data add to the growing literature pointing to a role for the TMA/TMAO pathway in olfaction and neurobehavior.

---

## [Author Response]

The following is the authors’ response to the original reviews.

**Reviewer #1 (Public review):**
Summary:This study focuses on the bacterial metabolite TMA, generated from dietary choline. These authors and others have previously generated foundational knowledge about the TMA metabolite TMAO, and its role in metabolic disease. This study extends those findings to test whether TMAO's precursor, TMA, and its receptor TAAR5 are also involved and necessary for some of these metabolic phenotypes. They find that mice lacking the host TMA receptor (Taar5-/-) have altered circadian rhythms in gene expression, metabolic hormones, gut microbiome composition, and olfactory and innate behavior. In parallel, mice lacking bacterial TMA production or host TMA oxidation have altered circadian rhythms.Strengths:These authors use state-of-the-art bacterial and murine genetics to dissect the roles of TMA, TMAO, and their receptor in various metabolic outcomes (primarily measuring plasma and tissue cytokine/gene expression). They also follow a unique and unexpected behavioral/olfactory phenotype. Statistics are impeccable.Weaknesses:Enthusiasm for the manuscript is dampened by some ambiguous writing and the presentation of ideas in the introduction, both of which could easily be improved upon revision.

We apologize for the abbreviated and ambiguous writing style in our original submission. Given Reviewer 2 also suggested reorganizing and rewriting certain parts, we have spent time to remove ambiguity by adding additional points of clarification and adding more historical context to justify studying TMA-TAAR5 signaling in regulating host circadian rhythms. We have also reorganized the presentation of data aligned with this.

**Reviewer #2 (Public review):**
Summary:In the manuscript by Mahen et al., entitled "Gut Microbe-Derived Trimethylamine Shapes Circadian Rhythms Through the Host Receptor TAAR5," the authors investigate the interplay between a host G protein-coupled receptor (TAAR5), the gut microbiota-derived metabolite trimethylamine (TMA), and the host circadian system. Using a combination of genetically engineered mouse and bacterial models, the study demonstrates a link between microbial signaling and circadian regulation, particularly through effects observed in the olfactory system. Overall, this manuscript presents a novel and valuable contribution to our understanding of hostmicrobe interactions and circadian biology. However, several sections would benefit from improved clarity, organization, and mechanistic depth to fully support the authors' conclusions.Strengths:(1) The manuscript addresses an important and timely topic in host-microbe communication and circadian biology.(2) The studies employ multiple complementary models, e.g., Taar5 knockout mice, microbial mutants, which enhance the depth of the investigation.(3) The integration of behavioral, hormonal, microbial, and transcript-level data provides a multifaceted view of the observed phenotype.(4) The identification of olfactory-linked circadian changes in the context of gut microbes adds a novel perspective to the field.Weaknesses:While the manuscript presents compelling data, several weaknesses limit the clarity and strength of the conclusions.(1) The presentation of hormonal, cytokine, behavioral, and microbiome data would benefit from clearer organization, more detailed descriptions, and functional grouping to aid interpretation.

We appreciate this comment and have reorganized the data to improve functional grouping and readability. We have also added additional detail to descriptions of the data in the revised figure legends and results.

(2) Some transitions-particularly from behavioral to microbiome data-are abrupt and would benefit from better contextual framing.

We agree with this comment, and have added additional language to provide smoother transitions. This in many cases brings in historical context of why we focused on both behavioral and microbiome alterations in this body of work.

(3) The microbial rhythmicity analyses lack detail on methods and visualization, and the sequencing metadata (e.g., sample type, sex, method) are not clearly stated.

We apologize for this, and have now added more detail in our methods, figures, and figure legends to ensure the reader can easily understand sample type, sex, and the methods used.

(4) Several figures are difficult to interpret due to dense layouts or vague legends, and key metabolites and gene expression comparisons are either underexplained or not consistently assessed across models.

Aligned with the last comment we now added more detail in our methods, figures, and figure legends to provide clear information. We have now provided additional data showing the same key metabolites, hormones, and gene expression alterations in each model if the same endpoints were measured.

(5) Finally, while the authors suggest a causal role for TAAR5 and its ligand in circadian regulation, the current data remain correlative; mechanistic experiments or stronger disclaimers are needed to support these claims.

We agree with this comment, and as a result have removed any language causally linking TMA and TAAR5 together in circadian regulation. Instead, we only state finding in each model and refrain from overinterpreting.

**Reviewer #3 (Public review):**
Summary:Deletion of the TMA-sensor TAAR5 results in circadian alterations in gene expression, particularly in the olfactory bulb, plasma hormones, and neurobehaviors.Strengths:Genetic background was rigorously controlled.Comprehensive characterization.Weaknesses:The weaknesses identified by this reviewer are minor.Overall, the studies are very nicely done. However, despite careful experimentation, I note that even the controls vary considerably in their gene expression, etc, across time (eg, compare control graphs for Cry 1 in IB, 4B). It makes me wonder how inherently noisy these measurements are. While I think that the overall point that the Taar5 KO shows circadian changes is robust, future studies to dissect which changes are reproducible over the noise would be helpful.

We thank the reviewer for this insightful comment. We completely agree that there are clear differences in the circadian data in experiments from Taar5^-/-^ mice and those from gnotobiotic mice where we have genetically deleted CutC. Although the data from Taar5^-/-^ mice show nice robust circadian rhythms, the data from mice where microbial CutC is altered have inherently more “noise”. We attribute some of this to the fact that the Taar5^-/-^ mouse experiment have a fully intact and diverse gut microbiome . Whereas, the gnotobiotic study with CutC manipulation includes only a 6 member microbiome community that does not represent the normal microbiome diversity in the gut. This defined synthetic community was used as a rigorous reductionist approach, but likely affected the normal interactions between a complex intact gut microbiome and host circadian rhythms. We have added some additional discussion to indicate this in the limitations section of the manuscript.

Impact:These data add to the growing literature pointing to a role for the TMA/TMAO pathway in olfaction and neurobehavioral.
**Reviewer #1 (Recommendations for the authors):**
I suggest a revision of the writing and organization. The potential impact of the study after reading the introduction is unclear. One example, in the intro, " TMAO levels are associated with many human diseases including diverse forms of CVD5-12, obesity13,14, type 2 diabetes15,16, chronic kidney disease (CKD)17,18, neurodegenerative conditions including Parkinson's and Alzheimer's disease19,20, and several cancers21,22" It would be helpful to explain how the previous literature has distinguished that the driver of these phenotypes is TMA/TMAO and not increased choline intake. Basically, for a TMA/O novice reader, a more detailed intro would be helpful.

We appreciate this insightful comment and have now provided a more expansive historical context for the reader regarding the effects of choline consumption (which impacts many things, including choline, acetylcholine, phosphatidylcholine, TMA, TMAO, etc) versus the primary effects of TMA and TMAO.

There were also many uses of vague language (regulation/impact/etc). Directionality would be super helpful.

We thank the reviewer for this recommendation and have improved language as suggested to show directionality of our findings. The terms regulation, impact, shape etc. are used only when we describe multiple variable changing at the same time over the time course of a 24-hour circadian period (some increased and some decreased).

**Reviewer #2 (Recommendations for the authors):**
In the manuscript by Mahen et al., entitled "Gut Microbe-Derived Trimethylamine Shapes Circadian Rhythms Through the Host Receptor TAAR5," the authors investigate the interplay between a host G protein-coupled receptor (TAAR5), the gut microbiota-derived metabolite trimethylamine (TMA), and the host circadian system. Using a combination of genetically engineered mouse and bacterial models, the study demonstrates a link between microbial signaling and circadian regulation, particularly through effects observed in the olfactory system. Overall, this manuscript presents a novel and valuable contribution to our understanding of hostmicrobe interactions and circadian biology. However, several sections would benefit from improved clarity, organization, and mechanistic depth to fully support the authors' conclusions. Below are specific major and minor suggestions intended to enhance the presentation and interpretation of the data.Major suggestions:(1) Consider adding a schematic/model figure as Panel A early in the manuscript to help readers understand the experimental conditions and major comparisons being made.

We thank the reviewer for this recommendation and have added a graphical abstract figure to help the reader understand the major comparisons being made.

(2) Could the authors present body weight and food intake characteristics in Taar5 KO vs. WT animals?

We have added body weight data as requested in Figure 1, Figure supplement 1. Although we have not stressed these mice with a high fat diet for these behavioral studies, under chow-fed conditions studied here we did not find any significant differences in body weight. Given no difference in body weight, we did not collect data on food consumption and have mentioned this as a limitation in the discussion.

(3) Several figures, especially Figures 3 and 4, and Supplemental Figures, would benefit from more structured organization and expanded legends. Grouping related data into thematic panels (e.g., satiety vs. appetite hormones, behavioral domains) may help improve readability.

We appreciate the reviewer’s thoughtful comments and agree that reorganization would improve clarity. We have reorganized figures to improve clarity and have expanded the figure legends to provide more detail on experimental methods.

(4) Clarify and expand the description of hormonal and cytokine changes. For instance, the phrase "altered rhythmic levels" is vague - do the authors mean dampened, phase-shifted, enhanced, etc., relative to WT controls?

Given a similar suggestion was made by Reviewer 1, we have provided more precise language focused on directionality and which specific endpoints we are referring to. For anything looking at circadian rhythms, the revised manuscript includes specific indications when we are discussing mesor, amplitude, and acrophase alterations. The terms regulation, impact, shape etc. are used only when we describe multiple complex variables changing at the same time over the time course of a 24-hour circadian period (some increased and some decreased).

(5) Consider grouping hormones and cytokines functionally (e.g., satiety vs. appetite-stimulating, pro- vs. antiinflammatory) to better interpret how these changes relate to the KO phenotype.

We thank the reviewer for this recommendation, and have re-organized figure panels to reflect this.

(6) Please provide a more detailed description of the behavioral results, particularly those in Supplemental Figure 2.

We have both expanded the methods description in the revised figure legends, but have also added a more detailed description of the behavioral results.

(7) As with hormonal data, behavioral outcomes would be easier to follow if organized thematically (e.g., locomotor activity, anxiety-like behavior, circadian-related behavior), especially for readers less familiar with behavioral assays.

We appreciate this reviewer’s comment and agree that we can better group our data to show how each test is associated with the type of behavior it assesses. As a result we have reorganized the behavioral data into broad categories such as olfactory-related, innate, cognitive, depressive/anxiety-like, or social behaviors. We have also new data in each of these behavioral categories to provide a more comprehensive understanding of behavioral alterations seen in Taar5^-/-^ mice.

(8) The following statement needs clarification: "Also, it is important to note that many behavioral phenotypes examined, including tests not shown, were unaltered in Taar5-/- mice (Figures S2G, S2H, and S2I)." Consider rephrasing to explicitly state the intended message: are the authors emphasizing a lack of behavioral phenotype, or highlighting specific unaltered aspects?

We apologize for this confusing statement, and have changed the verbiage to improve readability. To expand the comprehensive nature of this study, we also now include the tests that were “not shown” in the original submission to provide a more comprehensive understanding of behavioral alterations seen in Taar5^-/-^ mice. These new data are included as 6 different figure supplements to main Figure 2.

(9) The transition from behavior to microbiome data feels abrupt. Can the authors better explain whether the behavioral changes are thought to result from gut microbial function, independent of TMA-Taar5 signaling?

We apologize for the poor transitions in our writing style. We have spent time to explain the previous findings linking the TMA pathway to circadian reorganization of the gut microbiome (mostly coming from our original paper Schugar R, et al. 2022, eLife) and how this correlates with behavioral phenotypes. Although at this point it is difficult to know whether the microbiome changes are driving behavioral changes, or vice versa it could be central TAAR5 signaling is altering oscillations in gut microbiome, we present our findings here as a framework for follow up studies to more precisely get at these questions. It is important to note that our experiment using defined community gnotobiotic mice with or without the capacity to produce TMA (i.e. CutC-null community) shows that clearly microbial TMA production can impact host circadian rhythms in the olfactory bulb. Additional experiments beyond the scope of this work will be required to test which phenotypes originate from TMA-TAAR5 signaling versus more broad effects of the restructured gut microbiome.

(10) For Figure 3A, please expand the microbiome results with more granularity:(a) Indicate in the Results section whether the sequencing method was 16S amplicon or metagenomic.

Sequencing was done using 16S rRNA amplicon sequencing using methods published by our group (PMID: 36417437, PMID: 35448550).

(b) State whether samples were from males, females, or a mix.

We have indicated that all mice from Figure 1 were male mice in the revised figure legend.

(c) Clarify whether beta diversity is based on phylogenetic or non-phylogenetic metrics. Consider using both types if not already done.

Beta diversity was analyzed using the Bray-Curtis dissimilarity index as the metric. Details have been included in the methods section.

(d) Make lines partially transparent in the Beta-diversity plot so that individual points are visible.

We have now updated the Beta-diversity plot with individual points visualized.

(e) Clarify what percentage of variation in the Beta-diversity plot is explained by CCA1, and whether this low percentage suggests minimal community-level differences.

We have updated the Beta-diversity plot to include the R^2^ and p-values associated with these data.

(f) Confirm if the y-axis on the Beta-diversity plot should be labeled CCA2 rather than "CCAA 1".

We appreciate this comments, given it identified a typographical error in the plot. The revised figure now include the proper label of CCA2 instead of CCAA 1.

(11) For Figure 3B:(a) Provide a description of the taxonomy plot in the results.

We have added a description of the taxonomy plot in the revised results section.

(b) Add phylum-level labels and enlarge the legend to improve the readability of genus-level data.

We agree this is a good suggestion so have enlarged the legend for the genus-level data and have also added phylum-level plots as well in the revised manuscript in Figure 3, figure supplement 1.

(12) Rhythmicity of the microbiome is central to the manuscript. The current approach of comparing relative abundance at discrete time points is limiting.

We thank the reviewer for this comment. We agree with this statement that discrete timepoint are not enough to describe circadian rhythmicity. In addition to comparing genotypes at discrete time points, we also used a rigorous cosinor analysis to plot the data over a 24-hour time period, and those differences are shown in the figure itself as well as Table 1.

(a) Please describe how rhythmicity was determined, e.g., what data or statistical method supports the statement: "Taar5-/- mice showed loss of the normal rhythmicity for Dubosiella and Odoribacter genera yet gained in amplitude of rhythmicity for Bacteroides genera (Figure 3 and S3)."

We appreciate this reviewer comment. Rhythmicity was determined using a cosinor analysis by use of an R program. Cosinor analysis is a statistical method used to model and analyze rhythmic patterns in time-series data, typically assuming a sinusoidal (cosine) shape. It estimates key parameters like mesor (mean level), amplitude (height of oscillation), and acrophase (timing of the peak), making it especially useful in fields like chronobiology and circadian rhythm research. We have used this in previous research to describe circadian rhythms. We do plan to improve language considering directionality of these circadian changes.

(b) Supplemental Figure S3 needs reorganization to highlight key findings. It's not currently clear how taxa are arranged or what trends are being shown.

The data in Figure S3 show the entire 24-hour time course of the cecal taxa that were significantly altered for at least one time point between Taar5^+/+^ and Taar5^-/-^ mice. Given we showed time pointspecific alterations in the Main Figure 3, we thought these more expansive plots would be important to show to depict how the circadian rhythms were altered.

(c) Supplemental Table 1, which includes 16S features, should be referenced and discussed in the microbiome section.

We have now referenced and discussed Supplemental Table 1 which includes all cosinor statistics for microbiome and other data presented in circadian time point studies.

(13) Did the authors quantify the 16S rRNA gene via RT-PCR to determine if this was similar between KO and WT over the 24-hour period?

We did not quantify 16S rRNA gene via RT-PCR, but do not think adding this will change our overall interpretations.

(14) Reorganize Figure 4 to align with the order of results discussed-starting with TMA and TMAO, followed by related metabolites like choline, L-carnitine, and gamma-butyrobetaine.

We thank the reviewer for this comment. We have chosen this organization because it is ordered from substrates (choline, L-carnitine, and betaine) to the microbe-associated products (TMA then TMAO). We will improve the writing associated with this figure to clearly explain this organization.

(a) Although the changes in the latter metabolites are more modest, they may still have physiological relevance. Could the authors comment on their significance?

We appreciate this reviewer comment and agree. We have expanded the results and discussion to address this.

(15) The authors note similarities in circadian gene expression between Taar5 KO mice and Clostridium sporogenes WT vs. ΔcutC mice, but the gene patterns are not consistent.(a) Can the authors clarify what conclusions can reasonably be drawn from this comparison?

We hesitate to make definitive conclusions in the manuscript on why the gene patterns are not consistent, because it would be speculation. However, one major factor likely driving differences is the status of the diversity of the gut microbiome in the different studies. For instance, in the studies using Taar5^+/+^ and Taar5^-/-^ mice there is a very diverse microbiome in these conventionally housed mice. In contrast, by design the experiment using Clostridium sporogenes WT vs. ΔcutC communities is a reductionist approach that allows us to genetically define TMA production. In these gnotobiotic mice, the simplified community has very limited diversity and this likely alters the host circadian rhythms in gene expression quite dramatically. Although it is impossible to directly compare the results between these experiments given the difference microbiome diversity, there are clearly alterations in host gene expression when we manipulate TMA production (i.e. ΔcutC community) or TMA sensing (i.e. Taar5^-/-^).

(16) Were circadian and metabolic genes (e.g., Arntl, Cry1, Per2, Pemt, Pdk4) also analyzed in brown adipose tissue of Taar5 KO mice, and how do these results compare to the Clostridium models?

We thank the reviewer for this comment. Unfortunately, we did not collect brown adipose tissue in our original Taar5 study. We plan on doing this in future follow up studies studying cold-induced thermogenesis that are beyond the scope of this manuscript. However, we have decided to include data from our two timepoint Taar5 study which looks at ZT2 (9am) and ZT14 (9pm). There are clear differences in circadian genes between these timepoints.

(17) To allow a more direct comparison, please ensure the same cytokines (e.g., IL-1β, IL-2, TNF-α, IFN-γ, IL6, IL-33) are reported for both the Taar5 KO and microbial models.

We thank the reviewer for this comment and now include data from the same cytokines for each study.

(18) What was the defined microbial community used to colonize germ-free mice with C. sporogenes strains? Did this community exhibit oscillatory behavior?

To define TMA levels using a genetically-tractable model of a defined microbial community, we leveraged access to the community originally described by our collaborator Dr. Federico Rey (University of Wisconsin – Madison) (PMID: 25784704). We chose this community because it provide some functional metabolic diversity and is well known to allow for sufficient versus deficient TMA production. We are thankful for the reviewer comments about oscillatory behavior of this defined community, and to be responsive have performed sequencing to detect the species over time. These data are now included in the revised manuscript and show that there are clear differences in the oscillatory behavior of the defined community members. These data provide additional support that bacterial TMA production not only alters host circadian rhythms, but also the rhythmic behavior of gut bacteria themselves which has never been described before.

(19) Can the authors explain the rationale for measuring additional metabolites such as tryptophan, indole acetic acid, phenylacetic acid, and phenylacetylglycine? How are these linked to CutC gene function or Taar5 signaling?

We appreciate that this could be confusing, but have included other gut microbial metabolites to be as comprehensive as possible. This is important to include because we have found in other gnotobiotic studies where we have genetically altered metabolite production, if we alter one gut microbe-derived metabolite there can be unexpected alterations in other distinct classes of microbe-derived metabolites (PMID: 37352836). This is likely due to the fact that complex microbe-microbe and microbehost interactions work together to define systemic levels of circulating metabolites, influencing both the production and turnover of distinct and unrelated metabolites.

(20) The authors make several strong claims suggesting that loss of Taar5 or disruption of its ligand directly alters the circadian gene network. However, the current data are correlative. The authors should clarify that these findings demonstrate associations rather than direct causal effects, unless additional mechanistic evidence is provided. Approaches such as studies conducted in constant darkness, measurements of wheelrunning behavior, or analyses that control for potential confounding factors, e.g., inflammation or metabolic disruption, would help establish whether the observed changes in clock gene expression are primary or secondary effects. The authors are encouraged to either soften these causal claims or acknowledge this limitation explicitly in the discussion.

We thank the reviewer for this comment. We agree and have softened our language about direct effects of TMA via TAAR5 because we agree the data presented here are correlative only.

Minor suggestions:(1) Avoid repetitive phrases such as "it is important to note..." for improved flow. Rephrasing these instances will enhance readability.

We thank the reviewer for this suggestion and have deleted such repetitive phrases.

(2) For Figure 2, remove interpretations above he graphs and use simple, descriptive panel labels, similar to those in Supplemental Figure 2.

We have removed these interpretations as suggested, but have retained descriptive panel labels to help the reader understand what type of data are being presented.

**Reviewer #3 (Recommendations for the authors):**
Minor:In Figure 1D, UCP1 does not appear to be significantly changed.

We thank the reviewer for this comment and agree that UCP1 gene expression is not significantly altered . However, given the key role that UCP1 plays in white adipose tissue beiging, which is suppressed by the TMAO pathway, we think it is critical to show that this effect appears unaffected by perturbed TMA-TAAR5 signaling.

It would be helpful, in the discussion, to summarize any consistent changes across Taar5 KO, CutC deletion, and FMO3 deletion.

We have added this to the discussion, but as discussed above we hesitate to make strong interpretations about consistency between the models because the microbiome diversity is so different between the studies, and we did not measure all endpoints in both models.

For the Cosinor analysis, it may be helpful to remove the p-values that are >0.05 from the figures.

We have now removed any non-significant p-values that are associated with our figures.

For Figure 2, Supplement 1E, what are the two bars for each genotype?

We appreciate the reviewer pointing this out and will further explain this test in the figure with labels and in the legend.